# Establishing mammalian GLUT kinetics and lipid composition influences in a reconstituted-liposome system

Albert Suades[1,4], Aziz Qureshi[1,4], Sarah E. McComas[1], Mathieu Coinçon[1], Axel Rudling[2], Yurie Chatzikyriakidou[1], Michael Landreh [3], Jens Carlsson [2] & David Drew [1] ✉

Glucose transporters (GLUTs) are essential for organism-wide glucose homeostasis in mammals, and their dysfunction is associated with numerous diseases, such as diabetes and cancer. Despite structural advances, transport assays using purified GLUTs have proven to be difficult to implement, hampering deeper mechanistic insights. Here, we have optimized a transport assay in liposomes for the fructose-specific isoform GLUT5. By combining lipidomic analysis with native MS and thermal-shift assays, we replicate the GLUT5 transport activities seen in crude lipids using a small number of synthetic lipids. We conclude that GLUT5 is only active under a specific range of membrane fluidity, and that *human* GLUT1-4 prefers a similar lipid composition to GLUT5. Although GLUT3 is designated as the high-affinity glucose transporter, in vitro D-glucose kinetics demonstrates that GLUT1 and GLUT3 actually have a similar $K_M$, but GLUT3 has a higher turnover. Interestingly, GLUT4 has a high $K_M$ for D-glucose and yet a very slow turnover, which may have evolved to ensure uptake regulation by insulin-dependent trafficking. Overall, we outline a much-needed transport assay for measuring GLUT kinetics and our analysis implies that high-levels of free fatty acid in membranes, as found in those suffering from metabolic disorders, could directly impair glucose uptake.

From molds to mammals, glucose is quantitatively the most important fuel source for life on Earth. To utilize glucose, the sugar must first be taken up into our cells by the action of glucose (GLUT) transporters, which despite low binding affinities ($K_d > 1$ mM), are capable of high specificity. In humans, there are 14 different GLUT isoforms, which differ in their substrate selectivity, tissue expression, and kinetics[1–4]. For example, *human* GLUT3 has an estimated $K_M$ for glucose at 1 mM to uptake glucose into neurons, whereas GLUT2 has a $K_M$ of around 18 mM and is essential for high turnover of glucose uptake into the liver after a meal[5]. GLUT5 is the only isoform thought to be specific to

fructose and is, after hydrolysis of sucrose, essential for the uptake of dietary fructose across the brush-border membrane of the small intestine. Various diseases are caused by impaired GLUT expression levels and transport activity, such as GLUT1 deficiency syndrome (De vivo disease)[6–13]. Moreover, due to the Warburg effect, GLUT expression is upregulated in various cancers[14–19].

Owing to its importance in metabolism, GLUTs are one of the most extensively studied types of solute carrier (SLC) transporters[1,3,20,21]. Due to the natural abundance of GLUT1 in erythrocytes and the availability of site-specific inhibitors, GLUT1 sugar

[1]Department of Biochemistry and Biophysics, Stockholm University, Svante Arrhenius v. 16c, SE–106 91 Stockholm, Sweden. [2]Science for Life Laboratory, Department of Cell and Molecular Biology, Uppsala University, BMC, Box 596, SE-751 24 Uppsala, Sweden. [3]Department of Microbiology, Tumor and Cell Biology, Karolinska Institutet, Solnavägen 9, SE-171 65 Solna, Sweden. [4]These authors contributed equally: Albert Suades, Aziz Qureshi. ✉e-mail: ddrew@dbb.su.se

specificity, and kinetics has been analyzed for several decades[21–24]. Other GLUT isoforms can be recombinantly expressed in *Xenopus* oocytes[25–27], which has proven a useful host for studying GLUT kinetics and assessing mutations. Crystal and cryo-EM structures of GLUT1[28–30], GLUT3[29,31], GLUT4[32], GLUT5[33], and the *Plasmodium falciparum* hexose transporter homolog (*Pf*HT1)[34] have provided a molecular framework for sugar recognition and transport, which is consistent with the biochemical analysis pre-dating structural information[5]. Interestingly, D-glucose is coordinated almost entirely by residues from the C-terminal domain[31]. Structures have shown how the half-helices TM7a–b and TM10a–b of the C-terminal bundle undergo local conformational changes to bind and release the sugar during the transport cycle[35].

Whilst we have a fairly good structural understanding of the GLUT transport cycle[35], in vitro GLUT transport assays have been unable to replicate the kinetic ($K_M$) estimates measured from in vivo GLUT kinetics[36,37]. The reason for the discrepancy might be due to the inherent instability of purified GLUT proteins[36]. Another major issue is that GLUT proteins are passive transporters (non-energized), and, as such, the uptake difference between liposomes reconstituted with and without protein can be low[33]. Establishing an in vitro GLUT transport assay is important since the relationship of GLUT activities and whole-cell D-glucose metabolism is based on a comparison of 2-deoxy-D-glucose uptake and not D-glucose uptake[5,26,27] due to the rapid in vivo phosphorylation of D-glucose. Another important kinetic parameter is transport turnover ($k_{cat}$), which is used to estimate enzyme catalytic efficiency ($k_{cat}/K_M$), but this is a difficult parameter to estimate in cells, as it requires an accurate quantification of the specific transporter in the cell membrane. Furthermore, how individual lipids influence GLUT activity needs to be more thoroughly investigated, as in previous studies of *human* GLUT3 and GLUT4 in proteoliposomes the $K_M$ estimates for D-glucose were 10-fold higher than what was expected by in vivo studies[5,37], i.e., it is unclear if the interpretation of lipid preferences uncovered would be relevant under more optimal transport conditions. Lastly, in the fructose transporter GLUT5, proteoliposome assays have been used to evaluate inhibitors and mutations[38,39], but since kinetic parameters were not reported, the conclusions drawn from this analysis are in need of further validation. For these reasons, we set out to establish a proteoliposome assay for the fructose transporter GLUT5, which could then be used to analyze the role of individual lipids and the kinetics of other GLUT transporter isoforms.

## Results

### Selection of crude lipids for proteoliposome GLUT transport assays

Previously, we found that purified *human* GLUT1 could be reconstituted by the rapid dilution and freeze/thaw method into liver lipids supplemented with cholesteryl hemisuccinate (CHS) to give a high signal-to-noise for D-glucose transport[33]. Unfortunately, this setup gave a low signal-to-noise for the uptake of D-fructose by reconstituted *rat* GLUT5 (Supplementary Fig. 1a), which we partly attributed to the reported 10-fold poorer binding affinity of GLUT5 for D-fructose[19]. Due to the high background signal from the non-specific uptake of $^{14}$C-D-fructose, analysis of GLUT5 variants was not possible. After extensive optimization of the GLUT5 proteoliposome assay, we were still unable to improve specific activity. We also encountered lipid-batch inconsistencies, with the final liver-lipid batches giving no detectable D-fructose transport altogether (Fig. 1a). Therefore, we decided to screen for GLUT5 transport using liposomes made from other crude sources, starting with *Escherichia coli* and *Soya* lipids plus CHS respectively, which had previously been used in GLUT3 and GLUT5 proteoliposome assays[31,39]. Similar to previous studies[31,39]. However, we found that the signal-to-noise remained poor at ~3:1, using the rapid-dilution method (Fig. 1a). We next tested GLUT5 activity using liposomes made from commercially available mammalian lipid extracts plus CHS. Strikingly, we observed that using liposomes made from *bovine* brain-fraction-

seven increased the specific activity of D-fructose transport by GLUT5, with an overall signal-to-noise of ~10:1 (Fig. 1a).

### Establishing GLUT5 fructose kinetics with liposomes generated from brain lipids

In the rapid-dilution method[40], a very high lipid-to-protein ratio is used to ensure an overabundance of lipids for the protein to spontaneously reconstitute into. It is possible, however, that a lower lipid-to-protein ratio might improve activity. We assayed transport activity after changing lipid concentration, but using fewer lipids had a negative impact on GLUT5 activity (Supplementary Fig. 1b). Screening for GLUT5 activity upon varying the amount of added protein from 5 to 30 µg, we observed that protein reconstitution efficiency increased linearly with the amount of GLUT5 (Supplementary Fig. 1c). Respective transport assays were repeated and likewise showed a linear increase in GLUT5 transport activity at both 30 s and 2 min time intervals (Supplementary Fig. 1d). We concluded that 20 µg of GLUT5 protein and 30 mg/mL lipids were optimal amounts to achieve a robust signal and to avoid protein saturation. With these parameters, the reconstitution efficiency was estimated at ~30–40%, with no major differences in the reconstitution efficiency between liposomes made from either brain lipids or any other of the crude lipids tested (Fig. 1b). Protein reconstitution was calculated by solubilizing the proteoliposomes with DDM (see methods for details). Since transport differences were not due to differences in protein reconstitution levels, we wondered if the higher GLUT5 activity observed in liposomes made from brain lipids was caused by lipid-dependent stabilization of the transporter. To test this, we measured the change in GLUT5 thermostability upon the addition of the different crude lipids by GFP-TS[41,42] but found that GLUT5 stability in brain-fraction-seven lipids was either lower or similar to the other lipid mixtures tested, including those from *E. coli* (Fig. 1c, d). Rather, GLUT5 was most stable in brain-fraction-1 lipids, which showed poor GLUT5 activity (Fig. 1a).

Using optimized brain-fraction-seven proteoliposomes, a robust time-course for D-fructose by GLUT5 could be obtained (Fig. 1e). Michaelis–Menten zero-trans kinetics of GLUT5 were determined, with a $K_M$ of 11 mM and a $k_{cat}$ of 43 s$^{-1}$ (Fig. 1f), which matched previously reported $K_M$ value of 8–11 mM from oocytes[43,44], and the reported $K_d$ of 9.2 mM for D-fructose binding determined using tryptophan-fluorescence quenching[33]. The observed turnover by *rat* GLUT5, at 43 molecules per second, is similar to the $k_{cat}$ of 28 s$^{-1}$ lactose uptake by LacY in proteoliposomes[45]. Based on these kinetic parameters and the high signal-to-noise in the time-course experiments, we conclude that the in vitro proteoliposome set-up reported here is able to recapitulate in vivo GLUT5 kinetics for D-fructose.

### Probing GLUT5 sugar specificity and ligand interactions

GLUT5 is thought to be the only GLUT transporter specific for D-fructose, which is predominantly expressed in the small intestine performing dietary fructose uptake[44,46]. To confirm the in vivo-based classification of GLUT5, we measured $^{14}$C-D-fructose uptake in competition with a number of D-fructose epimers and other relevant non-isotope-labeled sugars at ~2000-fold molar excess (see "Methods"). As shown in Fig. 2a, b, only D-fructose and 2,5-anhydro-D-mannitol (2,5-AHM) reduced the uptake of $^{14}$C-D-fructose. 2,5-AHM is very similar to D-fructose but lacks an OH-group in the C2 position (Supplementary Fig. 2a). Neither of the D-fructose epimers which differ in the orientation on one of the other five OH-groups could compete for D-fructose uptake (Supplementary Fig. 2a, b). Their inability to compete confirms that the exact OH group orientation in all positions, except for the C2 position, is essential for GLUT5 to recognize D-fructose. Our analysis is entirely consistent with the strict substrate specificity of *human* GLUT5 as measured in Chinese hamster ovary (CHO) cells, including the ability to transport 2,5-AHM[47]. The sugar specificity analysis provides further support for the robustness of the GLUT5 proteoliposome assay

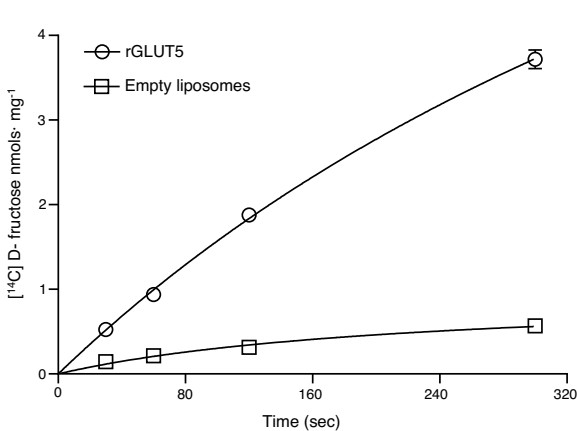

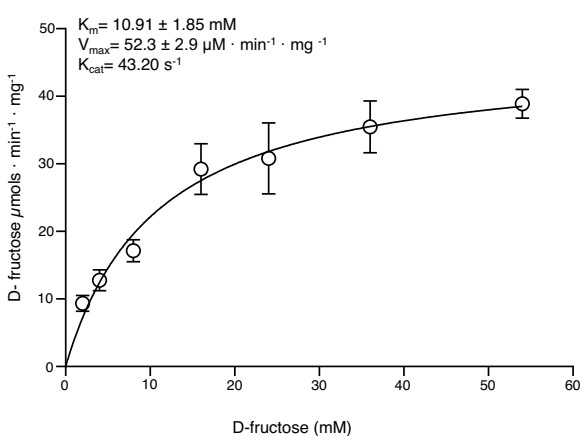

and corroborates that GLUT5 is strictly selective for D-fructose, despite its weak -10 mM binding affinity.

Having established an in vitro proteoliposome assay for measuring D-fructose transport by GLUT5, we proceeded to analyze mutants previously proposed to change the sugar specificity from D-fructose to D-glucose[33,38]. The closest GLUT5 isoform is GLUT7[33], which can transport both D-glucose and D-fructose. The only difference in the

binding pocket between these two isoforms is that the residue Gln166 in GLUT5 is glutamate in GLUT7[33]. Binding studies by tryptophan fluorescence spectroscopy demonstrated that a Gln166Glu mutation of GLUT5 was sufficient to shift the binding preference from D-fructose to D-glucose[33]. However, in our optimized transport assay, the Gln166Glu mutant is still unable to transport either D-fructose or D-glucose (Fig. 2c). We think this is unlikely to be a discrepancy of the

**Fig. 1 | Optimization of liposome-based transport assay for rGLUT5.**
**a** Comparison of lipid extract effect on liposome reconstituted rGLUT5 $^{14}$C-D-fructose uptake, at 2 min. rGLUT5 specific uptake and non-specific uptake into empty liposomes are shown as bars, colored in black and white. Errors indicated per bar represent s.e.m. of three independent experiments. **b** Reconstitution efficiency of rGLUT5 into liposomes prepared using different lipid extracts: Brain 7, Brain 1, *E. coli*, Liver, Soya total and Soya PC (see "Methods"). The y-axis indicates the relative band densitometry of rGLUT5, and each black bar represents a relative densitometry band of rGLUT5 for each lipid extract used for reconstitution as shown in Supplementary Fig. 1e; error bars show the range of two independent reconstitutions. **c** Melting curve of rGLUT5 using lipid extracts as shown in Fig. 1a. As a control, the melting temperature of purified rGLUT5 with DDM added, but no lipid is shown in black. The data are normalized fluorescence mean ± s.e.m. of $n = 3$ independent experiments. **d** The melting temperatures ($T_M$) for rGLUT5 in the presence of different lipid extracts, as calculated from the curves in panel (**c**). Error bars indicate mean ± s.e.m. of $n = 3$ independent experiments. **e** Time course uptake of $^{14}$C-D-fructose by rGLUT5 in proteoliposomes and empty liposomes made using brain-fraction-seven represented as empty circles and squares, respectively. Errors bars represent s.e.m. of three independent experiments. **f** Zero *trans* kinetics of rGLUT5 D-fructose transport using brain-fraction-seven proteoliposomes. All data points at their respective D-fructose concentration were measured after 50 s, and data were fitted using a non-linear Michaelis–Menten function by GraphPad prism. Error bars indicate mean ± s.e.m. of $n = 3$ independent experiments.

Trp-fluorescence measurements since the Ala395Gly mutant displayed wildtype D-fructose binding by tryptophan quenching[33] also retained high D-fructose transport (Fig. 2c). It is possible that the Gln166Glu mutant is able to bind D-glucose, but is not able to catalyze transport. The GLUT5 variant Ala395Trp, constructed to match the GLUT1 binding site, was also reported capable of D-glucose transport in proteoliposome assays[38]. However, the Ala395Trp mutant also abolished both D-fructose and D-glucose transport (Fig. 2c). Lastly, a conservative valine-to-isoleucine substitution has been reported to selectively abolish D-fructose transport in GLUT7[48], but the corresponding mutant in GLUT5 Val292Ile retained close to wildtype transport activity (Fig. 2c). Taken together, we are unable to confirm previously reported mutations altering sugar preferences of GLUT5. Rather, the directed evolution of hexose transporters and the analysis of the promiscuous sugar transporter from the malarial parasite *Pf*HT1 indicates that the extracellular gating regions connected to the sugar-binding site might also need to be mutated to alter sugar specificity[34,49], which is an avenue that can now be explored in greater depth with the optimized proteoliposome assay.

GLUT5 is highly expressed in breast cancer cells[19,50] and GLUT5 inhibitors have been developed, such as N-[4-(methanesulfonyl)-2-nitrophenyl]-1,3-benzodioxol-5-amine (MSNBA) (Supplementary Fig. 3a)[39,51]. We investigated D-fructose uptake by GLUT5 in the presence of 0.2 mM of MSNBA but found no competition for transport (Fig. 2d). As expected, the GLUT1-4 inhibitor cytochalasin B[39] also failed to inhibit GLUT5 D-fructose uptake (Fig. 2d). In an effort to develop GLUT5-specific inhibitors, we used molecular docking to screen a library with 1.6 million commercially available molecules against a crystal structure, and evaluated inhibition for a small subset of top-scoring compounds (see Methods). We uncovered one compound phenyl-n-(p-tolyl)-carbamate (C2), causing a 20% reduction in D-fructose transport by GLUT5 (Fig. 2d and supplementary Fig. 3a, b). We next trialed commercially available analogs and found one compound 4-methylphenyl-N-(4-methylphenyl)-carbamate (H4) with an improved inhibition of 40% (Fig. 2e and supplementary Fig. 3a). The IC$_{50}$ of compound H4 is a modest 0.5 mM, yet a large improvement in comparison to the affinity of either D-fructose at 10 mM or MSNBA, which shows no inhibition (Supplementary Fig. 3c). The H4 compound shows twofold selectivity toward GLUT5 as compared to GLUT1 (Supplementary Fig. 3c). Although clearly, the GLUT5 inhibitors will require extensive optimization, the analysis confirms that our proteoliposome setup is sensitive enough to detect small molecule inhibition, and highlights the importance of robust in vitro assays for validation.

**Elucidating the problems encountered with using suboptimal lipids for proteoliposome assays**
Concerned by the differences between the analysis carried out here and previous GLUT5 mutagenesis and inhibition studies[38,39], we endeavored to investigate the underlying reasons for these discrepancies. The main difference with our optimized protocol is the use of liposomes made with brain lipids instead of soya bean lipids. With our reconstitution protocol, we find that signal-to-noise in liposomes made from either Soya PC or Soya extract lipids was low at ~3:1 (Fig. 1a). It is possible that this poor signal for specific activity has led to inaccurate conclusions. To explore this further, we re-tested GLUT5 sugar specificity and were surprised to observe the non-physiological uptake of $^{14}$C-D-glucose in liposomes made with soya PC lipids with a similar signal-to-noise ~3:1 as D-fructose (Fig. 3a). Given we had clearly confirmed the specificity of GLUT5 for D-fructose in brain lipids, these findings were puzzling. We noticed that there was a higher level of sugar accumulation in protein-free liposomes prepared using soya lipid extract compared to using brain lipid extracts. Soya lipids have a much higher fraction of unsaturated lipids (see "Methods"), and we wondered if the liposomes in the presence of the GLUT5 protein were becoming "leaky", which caused non-protein-mediated accumulation of D-glucose. To test this possibility, we monitored the uptake of $^{14}$C-D-glucose of the GLUT5 Ala395Trp mutant, which we confirmed was inactive in brain lipid liposomes, and indeed found the accumulation of D-glucose in the liposomes made from soya PC lipids (Fig. 3b), as was previously reported[38]. Lacking a high-affinity GLUT5 inhibitor to confirm artefactual uptake, we switched to monitor $^{14}$C-D-glucose uptake in the malarial hexose transporter *Pf*HT1 in the presence of a high-affinity inhibitor (IC$_{50}$ = 1 μM)[52]. As expected, in the presence of the inhibitor at 250 μM, there was no detectable uptake of $^{14}$C-D-glucose in *Pf*HT1 proteoliposomes made from brain-fraction-seven lipids (Fig. 3c). However, when using the soya PC lipids, we still observed a large amount of $^{14}$C-D-glucose accumulation (Fig. 3c), confirming that the uptake was clearly an artifact from leaky liposomes. Taken together, we conclude that proteoliposome integrity is weakened in the soya PC lipids to such an extent that it allows sugar to diffuse passively into the proteoliposomes. This assessment reinforces the importance of proteoliposome transport assay optimization and appropriate controls, not only for GLUT proteins but for transporters in general.

**GLUT5 does not strongly associate tightly with any specific lipid**
Lipidomic analysis was carried out on all the crude lipids to understand the preference of GLUT5 for liposomes made from brain-fraction-seven lipids. As summarized in Fig. 3d, the lipid composition varies considerably between most of the different crude lipid fractions. The most noticeable difference of the brain-fraction-seven lipids is that this crude lipid composition more closely matches the lipid composition and distribution of a typical mammalian plasma membrane[53–55]. This includes brain-fraction-one crude lipids, which lack the main plasma membrane lipid phosphatidylcholine (PC) and are instead made up of 50% phosphatidylserine (PS) lipids. Notably, in other MFS transporters, such a high concentration of negatively-charged lipid has been shown to arrest the transporter in an inward-facing conformation[56]. This explanation would make sense with the increased GLUT5 thermostability previously observed in brain-fraction-one lipids and the inability to transport D-fructose (Fig. 1a, c).

To better understand the lipid preferences of GLUT5, synthetic lipids representing each of the six individual lipid classes found in brain-fraction-seven were individually added to assess their respective

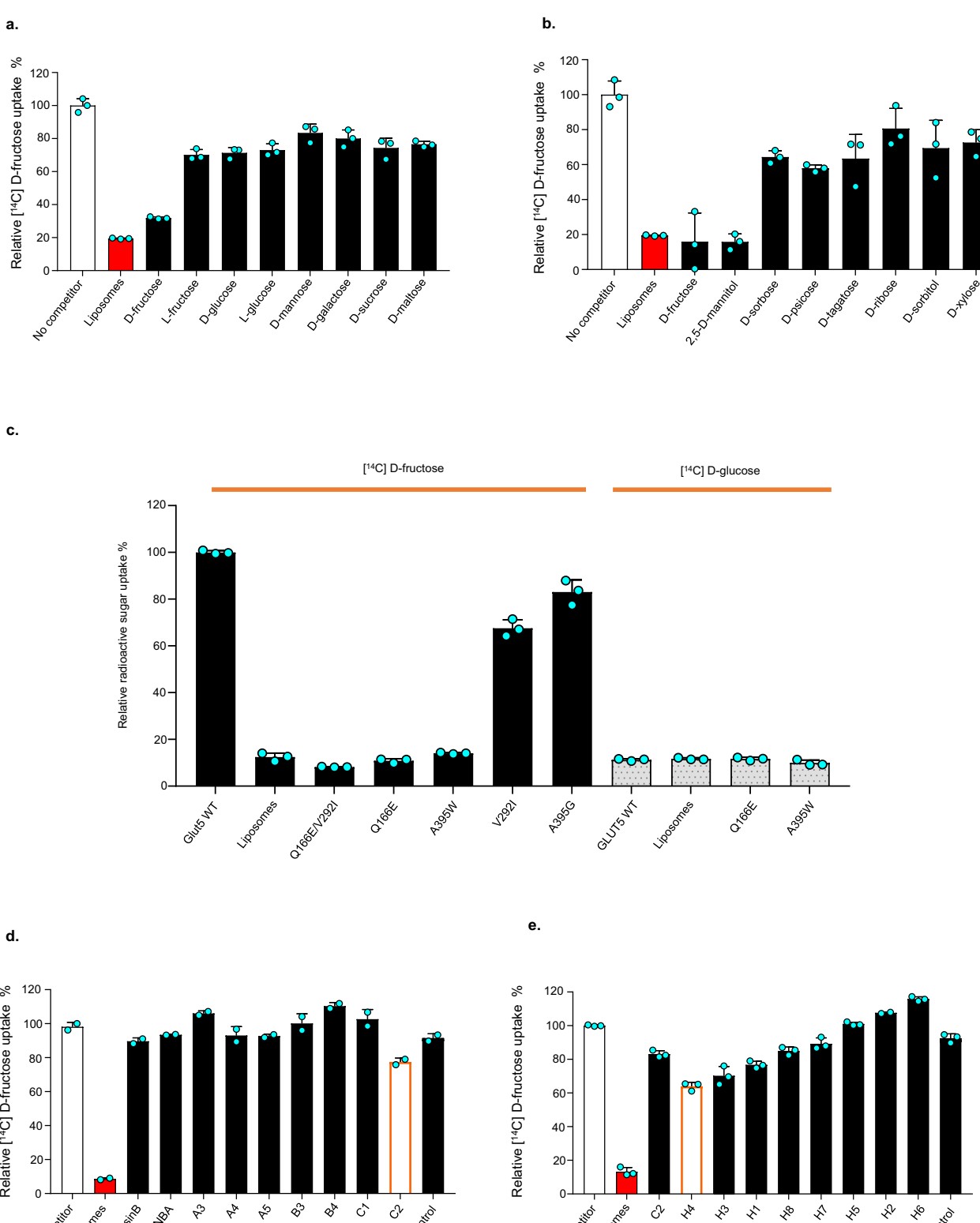

stabilization of detergent-purified GLUT5 using GFP-TS[41,42], namely, POPC, POPS, POPE, POPI, ceramide and sphingomyelin (Fig. 3e, f). As seen in Fig. 3e, most lipids demonstrated a modest increase in thermostabilization compared to DDM only ($\Delta T_M + 2-4\,°C$), with only POPI and POPS lipids driving a clear increase in thermostabilization ($\Delta T_M + 5\,°C$). To complement the protein stabilization by lipids, purified GLUT5 was analyzed by native MS (Supplementary Fig. 5a). The

spectrum for GLUT5 was well resolved, yet only a minor fraction of protein had lipid adducts, indicating no tightly bound lipids were retained during purification in detergent. Taken together, whilst high concentrations of PI and PS may bind to GLUT5, no particular lipid was found to bind strongly as seen by native MS, but it does corroborate the GLUT5 stabilization measured using brain-faction-one lipids (Fig. 1c, d).

**Fig. 2 | Characterization of rGLUT5 and inhibitor design. a** The competitive uptake of $^{14}$C-D-fructose by rGLUT5 in proteoliposomes prepared using brain-fraction-seven in the absence (white bar) and presence (black bars) of the non-labeled sugars with structures shown Supplementary Fig. 2a. Non-specific $^{14}$C-D-fructose uptake was estimated from decay activity recorded from liposomes without protein (red bar). Error bars indicate mean ± s.e.m. of $n = 3$ independent experiments. **b** As in **a**, with structures of non-labeled sugars shown in Supplementary Fig. 2b. **c** Uptake of $^{14}$C-D-fructose (black bars) and $^{14}$C-D-glucose (gray bars) by WT rGLUT5 and variants, in proteoliposomes prepared using brain-fraction-seven. Error bars indicate mean ± s.e.m. of $n = 3$ independent experiments. **d** Competitive uptake of $^{14}$C-D-fructose by rGLUT5 in proteoliposomes prepared

using brain-fraction-seven with some of the first-generation inhibitors at 100 μM, activity data for the other compounds are shown in supplementary Fig. 3b, structures of the respective inhibitors within this work in Supplementary Fig. 3a and Supplementary Fig. 4. Reported MSNBA inhibitor was included for comparison[39]. C2 compound competes mildly with $^{14}$C-D-fructose (empty bar with orange border). The structure of all compounds is shown in Supplementary Fig. 3a. Errors bars indicate the range of two independent experiments. **e** Commercially available derivatives of C2 were tested as in (**d**), and the most potent competitor for $^{14}$C-D-fructose is H2 (empty bar with orange border). The structure of all compounds is shown in Supplementary Fig. 3a. Error bars indicate mean ± s.e.m. of $n = 3$ independent experiments.

## Analyzing the impact of individual lipids by monitoring GLUT5 transport

Since brain-fraction-seven lipids do not seem to harbor any major lipids interacting strongly with GLUT5, the most rational explanation for the enhanced activity is that it has a more favorable lipid distribution. To understand what amounts of lipids are needed for optimal GLUT5 activity, individual synthetic lipids were combined at the same ratio as in brain-fraction-seven (see "Methods"). Using liposomes made from synthetic lipids, we were able to replicate D-fructose uptake to the level observed in liposomes made from brain-fraction-seven lipids (Fig. 4a). Having confirmed that the simplified liposome preparation was able to replicate the activity levels of the more complex brain-fraction-seven, we omitted each of the individual synthetic lipids during liposome preparation and re-analyzed D-fructose uptake (Fig. 4b). Somewhat unexpectedly, the omission of the major lipid PC (35%) had no impact on D-fructose transport, indicating the cylindrical sphingosine lipid SM (increased from 15% to 24%) could equally compensate the removal of these bilayer-forming glycerolphospholipids[57]. Indeed, the removal of SM lipids showed no real loss in activity when PC was present (Fig. 4b). The absence of POPS resulted in, on average, 10–20% reduced activity, which could be considered a larger effect since PS constitutes only 3.5% of the total brain-fraction-seven lipids (Fig. 4b, c). Likewise, the omission of POPI had a reduction in 20% of total transport, yet it is also present in minor amounts at 0.5%. Nevertheless, the most dramatic effect on activity was seen with the omission of cerebroside (30%) and POPE (15%), resulting in ~50–60% reduced transport activity, respectively. Phosphatidylethanolamine (PE) is a well-known and important non-bilayer-forming lipid associated with membrane protein curvature and fluidity. The lipid cerebroside is one the most common hexose-containing sphingosine lipids, which can also alter membrane bilayer fluidity by affecting the membrane phase. Taken together, while minor PI and PS lipids might be important to fine-tune activity through head-group interactions, it seems indirect effects by PE and cerebroside lipids might have a more critical role.

To better assess the importance of direct vs. indirect lipids, the POPS and POPE lipids were individually titrated against POPC lipids, and GLUT5 activity was re-measured (Fig. 4c, d). The incremental addition of 5% POPS lipids had a linear increase in transport activity, with a 50% increase in D-fructose uptake at a final concentration of 18% POPS lipids. This negatively-charged lipid comprises approximately 15% of total lipids in red blood cells and confirms the importance of PS lipid for GLUT5 transport, as was also concluded in proteoliposome studies of *human* GLUT3[37]. Increasing the PE lipid concentration also resulted in a clear increase of D-fructose uptake but with a 1.5-fold increase with a final concentration of 35% PE. Based on this analysis, it seems membrane curvature and fluidity might be just as important for GLUT5 transport as the presence of specific lipids like PS.

## Monitoring the effect of membrane fluidity on GLUT5 transport

PE is a conical-shaped lipid that introduces lipid-packing defects, affecting both membrane curvature and fluidity[58]. The PE headgroup has also been observed interacting directly with polar residues of

transporters during MD simulations, including XylE[35,59], but a direct PE-protein interaction has never been experimentally verified, making the impact of this lipid difficult to assess. In an effort to assess the importance of membrane fluidity more accurately, we decided to compare transport activity in liposomes made from POPC lipids with different degrees of acyl chain saturation. Acyl chains with a greater number of unsaturated double-bonds will increase membrane bilayer fluidity as the tails are more kinked and flexible. Remarkably, whilst we previously demonstrated that the omission of POPC (18:1) had no clear impact on transport activity (Fig. 4b), using PC lipids with two unsaturated acyl chains (18:2) dramatically increased activity to almost the same extent as the addition of the highest fraction of POPE/POPS lipids (Fig. 4c–e). Nonetheless, using PC lipids with six unsaturated bonds in the acyl chains (18:6) had a 75% reduction in transport activity (Fig. 4e), indicating an optimal level of fluidity between these degrees of unsaturation. To better estimate the impact of membrane fluidity, we carried out a titration between POPC (18:1) and DOPC (18:2) lipids. By increasing the concentration of DOPC, the net transport was increased more than twofold at 20% DOPC. We observed the transport rate was the highest with 20% DOPC but started to reduce again at 35% POPC, indicating that the maximal enhancement of GLUT5 activity is within a narrow range of membrane fluidity. Overall, we can confirm that membrane fluidity itself is an important contribution to GLUT5 transport.

## Assessing brain-fraction-seven lipids for other GLUT transporters

To evaluate the application of our optimized GLUT5 proteoliposome assay for other isoforms, *human* GLUT1, GLUT3, and GLUT4 transporters were expressed and purified from mammalian cells in a well-folded state (see "Methods" and Supplementary Fig. 5b–e). These GLUT transporters were forthwith reconstituted into all of the different crude lipids used for GLUT5 analysis, and the uptake of $^{14}$C-D-glucose was assessed. As a comparison, we included non-mammalian transporters; *Pf*HT1 from *Plasmodium falciparum* and the proton-coupled xylose transporter XylE from *E. coli*. For XylE, we monitored counterflow (passive) transport of D-xylose to eliminate complications by monitoring energized proton-coupled transport. As seen in Fig. 5a, the GLUTs and *Pf*HT1 both had the highest uptake in liposomes made from brain-fraction-seven lipids. The clear outlier was XylE, which, as expected, demonstrated its highest activity in liposomes made from it as the natural source (*E. coli* lipids). A time-course of GLUT1, GLUT3, and GLUT4 in liposomes made from brain-fraction-seven lipids demonstrated a good signal-to-noise for GLUT1, GLUT3, and GLUT4 around ~8:1 (Supplementary Fig. 5f–h). Since influx and efflux kinetics for GLUT proteins can be different[60], we further compared the orientation of the GLUT transporters in brain-fraction-seven liposomes. Essentially, GLUT–GFP fusion containing proteoliposomes was incubated with and without TEV, which will cleave off any GFP exposed to the outside. As shown in Supplementary Fig. 6, GLUT transporters are orientated with a similar ~70 to 80% biased orientation toward the physiological orientation. The main outlier is *Pf*HT1, which reconstitutes with only ~20% orientation to the physiological orientation,

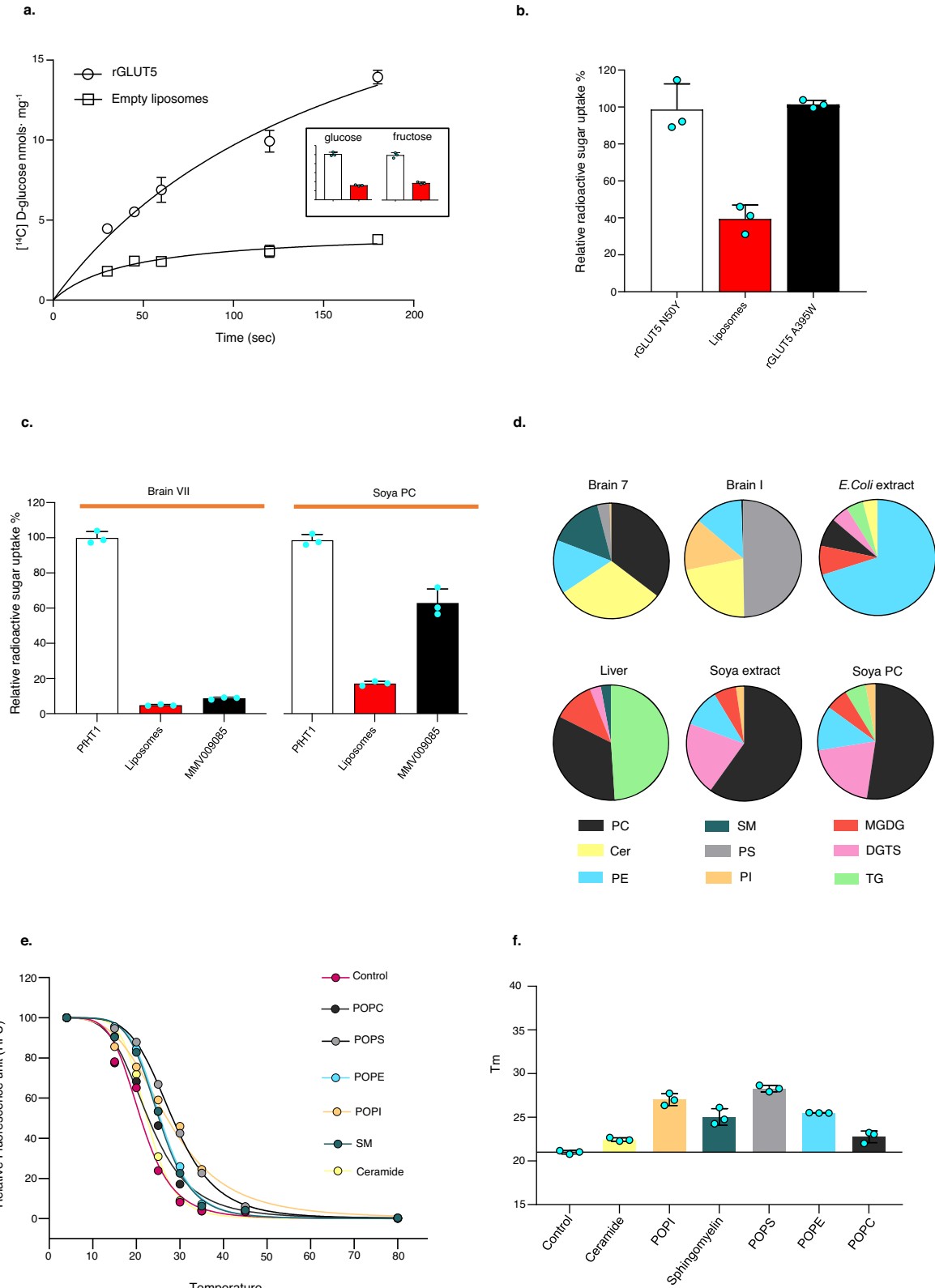

perhaps because this forms a detergent-stable homodimer[34]. Nevertheless, the $K_M$ for $Pf$HT1 in proteoliposomes matches the reported in vivo $K_M$ for D-glucose and D-fructose[34,61]; as such, orientation differences are unlikely to largely affect the interpretation of GLUT kinetics in this setup. We can thus compare GLUT1, GLUT3, GLUT4, and $Pf$HT1 zero-trans D-glucose kinetics under these optimized in vitro conditions (Fig. 5b–d and Supplementary Table 1).

While GLUT3 is often referred to as the high-affinity glucose transporter for neurons[62], the estimated $K_M$ for GLUT3 at 1.4 mM is only slightly lower than the $K_M$ of the ubiquitous GLUT1 transporter at 2.0 mM. Nonetheless, the $k_{cat}$ for GLUT3 at 12.5 s$^{-1}$ is 2-fold higher than the $k_{cat}$ of GLUT1 at 6.3 s$^{-1}$, which effectively means GLUT3 can more quickly uptake glucose into energetically demanding neurons. GLUT1 activity is saturated at 4 mM, which seems well placed for helping to

**Fig. 3 | Unspecific uptake of rGLUT5, lipidomic analysis of lipid extracts, and their respective stabilization. a** Uptake of $^{14}$C-D-glucose by rGLUT5 in proteoliposomes and empty liposomes prepared using soya PC represented as empty circles and squares, respectively. Inset; uptake of rGLUT5 (empty bars) and empty liposomes (red bars) at one-time point (2 min) in soya PC for glucose and fructose. Data were normalized to the uptake of rGLUT5 as 100%, and axis labels were not shown for clarity. Error bars indicate mean ± s.e.m. of $n = 3$ independent experiments. **b** Relative 2 min uptake of $^{14}$C-D-glucose by WT rGLUT5 (empty bars) and A395W (black bars) reconstituted into liposomes made from soya PC lipids. Non-specific uptake (red bars) into liposomes without protein is also represented. Data were normalized to WT rGLUT5 transport activity; error bars indicate mean ± s.e.m. of $n = 3$ independent experiments. **c** Uptake of $^{14}$C-D-glucose by PfHT1 (*Plasmodium falciparum* Hexose Transporter) in proteliposomes prepared using brain-fraction-seven and soya PC extracts, respectively, in the absence (empty bars) and presence (black bars) of inhibitor MMV009085 at 30 s. Non-specific $^{14}$C-D-Glucose uptake (red bars) is represented as in (**b**), and data was normalized to PfHT1 transport activity; error bars indicate mean ± s.e.m. of $n = 3$ independent experiments. **d** Lipidomics analysis of crude extracts from Fig. 1a. Phosphatidylcholine (PC) in black; ceramide (Cer) in light yellow; phosphatidylethanolamine (PE) in light blue; sphingomyelin (SM) in dark green; phosphatidylserine (PS) in light gray; phosphatidylinositol (PI) in light orange; monogalactosyldiacylglycerol (MGDG) in red; diacylglyceryltrimethylhomo-Ser (DGTS) in pink; and triglycerides (TG) in light green. **e** Melting curves of rGLUT5 in the presence of individual phospholipids, as colored in (**d**). As a control, the melting temperature of rGLUT5 with DDM added, but no lipid is shown in red; the data are normalized fluorescence mean ± s.e.m. of $n = 3$ independent experiments. **f** The melting temperature for rGLUT5 in the presence of individual phospholipids, as calculated from the curves in panel (**e**). Error bars indicate mean ± s.e.m. of $n = 3$ independent experiments.

maintain blood glucose levels around 5 mM[63]. The $K_M$ for GLUT4 is around 8.2 mM, which is consistent with the data reported on *Xenopus* oocytes[64] and more recent counterflow transport in proteoliposomes[32], but it has a low turnover of around 1 s$^{-1}$. It is interesting to mention that *human* GLUT4 has the lowest turnover for glucose despite having a higher $K_M$ than GLUT1 or GLUT3, which translates to the lowest specific activity ($k_{cat}/K_M$) for glucose (Fig. 5e). GLUT5 has a much lower affinity ($K_M$ at 11 mM) for D-fructose than the GLUT transporters for D-glucose, but with a much higher turnover of 43 s$^{-1}$. Given GLUT5 is also required to passively uptake D-fructose across the small intestine, where sugar concentrations after a sugary meal can be very high (>20 mM), these kinetic differences make sense. In contrast, D-glucose is transported across the small intestine by the sodium-coupled glucose transporter SGLT1[65], and therefore the GLUTs are only required to passively move the D-glucose into the bloodstream.

## Discussion

Available structures of different GLUT isoforms[28,29,31–33] and non-mammalian homologs in different conformations[34,66,67] has arguably completed the transport cycle of GLUT proteins[35]. Comparative analysis of available GLUT structures in different conformations with statistical approaches suggests a fairly "flat" free energy landscape[35]. Since GLUT proteins are uniporters, after releasing the sugar on one side of the membrane, the protein needs to go through an empty occluded state without a substrate to reset itself to face the other side, termed relaxation[35]. This is the rate-limiting step of the transport cycle, as it requires the protein to spontaneously reset itself in the absence of sugar. Indeed, one can speed up this step in radiolabeled-sugar influx assays by including cold sugar on the inside of liposomes so that the protein does not need to go through an empty occluded state, i.e., so-called trans-acceleration[4]. Interestingly, with decreasing temperature, the effect of trans-acceleration increases, which means that the spontaneous resetting becomes more rate-limiting as the temperature decreases[68]. Thus, GLUT protein dynamics itself might be governing transport.

Here, using carefully optimized GLUT5 transport assays in proteoliposomes, we prove that membrane fluidity is the explanation for this kinetic effect since, at lower temperatures, both membrane fluidity and protein dynamics are reduced. Notably, we have observed that there is an optimum window of membrane fluidity for GLUT5 transport, even in the presence of anionic lipids. Although negatively-charged lipids are clearly important[37], the native MS and thermal shift demonstrate that their influence is likely to be through transient head-group interactions rather than tight lipid-protein sites. Moreover, despite exhaustive efforts over many years, GLUT proteoliposomes assays have proven challenging to implement, implying that the experimental requirements needed were more than just the requirement of a specific lipid. Interestingly, the flat energy landscape of GLUTs suggests that incorrect lipid composition-fluidity might arrest the protein in one state or slow down the translocation cycle, hinting at why it has been so difficult to obtain a robust proteoliposome assay for monitoring sugar uptake. Notably, we demonstrate the importance of optimizing the in vitro transport assays for the interpretation of mutations and for inhibitor development. Lastly, we expect that the proteoliposome assay presented here will be of benefit to other mammalian SLC transporters, for which most currently lack workable in vitro transport assays for establishing their substrate-transporter pairing, inhibition, and energetics.

There are many structures now available of different GLUT isoforms and homologs[28,31–34,67,69], and although some structural differences are appreciated, like in PfHT1, the differences are minor, which reinforces that structural studies will not be sufficient to explain the functional differences between GLUT isoforms. To better understand the mechanism and dynamics of this family, it is fundamental to explore and compare their kinetic properties. Currently, there is a wealth of transport kinetic information on the different GLUT isoforms[26,29,64,70]. Nevertheless, most of the in vivo analyses show a large variation, making it difficult to compare kinetics between isoforms. Moreover, we are restricted to comparing D-glucose analogs, most often 2-deoxy-d-glucose (2-DG) or 3-O-methyl glucose (3-OMG), which are not natural substrates and therefore might show differential preferences towards certain GLUT isoforms than would D-glucose[26,29,71]. Obviously, in-vivo studies are limited to glucose analogs as D-glucose is quickly phosphorylated and metabolized, which is also the case in oocytes and red blood cells[64,71–73]. Nevertheless, in vivo, $K_m$ measurement of D-glucose analogs is a good starting point to estimate affinities, but it is difficult to evaluate $V_{max}$, $k_{cat}$, and their specific activity because it is complicated to calculate protein amounts present in cell membranes. Also, the vast majority of studies were carried out for one GLUT isoform at a time, with a few exceptions in oocytes[64], which introduces a further level of user bias. Lastly, *Xenopus* oocytes have a different membrane bilayer composition to mammalian plasma membranes; in particular, they have 10-fold less ceramide lipids[74], and modulating their lipid composition is not feasible. Indeed, as demonstrated here, GLUT proteins are very sensitive to the membrane bilayer composition.

Here, we have been able to estimate the kinetics of natural substrates D-glucose and D-fructose into proteoliposomes for GLUT1, GLUT3, GLUT4, and GLUT5 proteins. Whilst we cannot rule out a systematic difference between the absolute turnover of D-glucose into cells *versus* proteoliposomes, the $K_M$ estimates should not be affected, as has been shown to be the case for a number of unrelated transporters[75,76]. Out of the sugar transporters analyzed here, PfHT1 is the only transporter showing differences in orientation preferences and oligomerization, which could potentially lead to an under-estimation of its turnover ($k_{cat}$). Nevertheless, it serves as a useful comparison, and we can reliably compare the GLUT "relative" performance ($k_{cat}/K_M$). Here we have shown that GLUT1 and GLUT5 isoforms, showing the broadest tissue distribution for the specialized uptake of

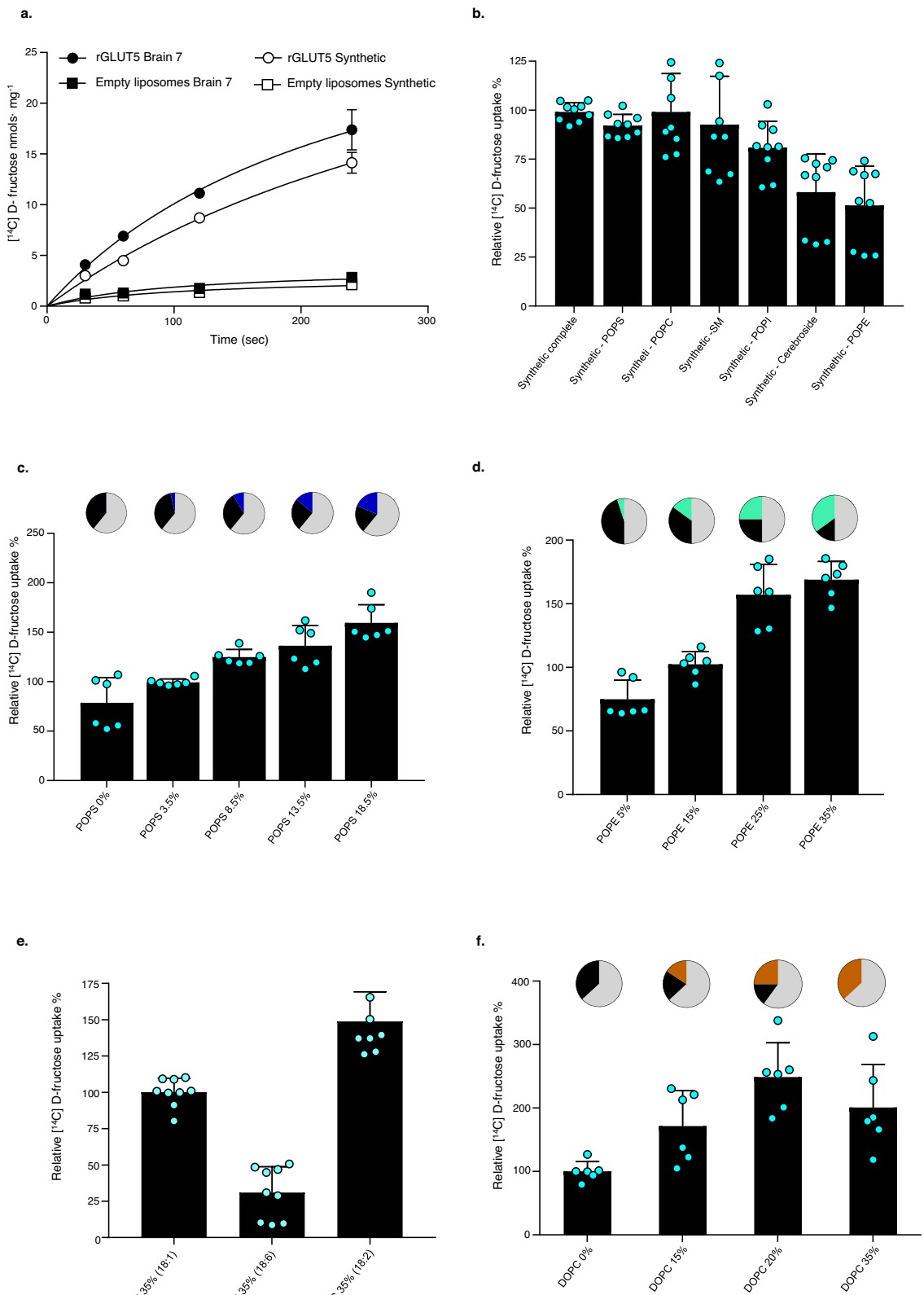

D-glucose and D-fructose sugars, respectively, operate at similar efficiency. Notably, whilst GLUT3 has been labeled as the high-affinity transporter, its ability to take up D-glucose into highly energy-demanding neurons is rather because it has a faster turnover. Lastly, the insulin-sensitive GLUT4 isoform has been reported to have a turnover similar to GLUT1[70]. However, our analysis demonstrates that GLUT4 is sixfold less efficient than GLUT1. We propose that since

GLUT4 transports D-glucose more slowly, its uptake will not compete with GLUT1, and instead, glucose uptake into fat and muscle cells can be fine-tuned by how much GLUT4 is present at the plasma membrane, i.e., in response to insulin.

Given the large complexity of mammalian membranes containing hundreds of different lipids, it's not straightforward to directly extrapolate our lipid analysis to the physiological situation. Nevertheless,

**Fig. 4 | Lipids influence the GLUT5 activity in proteoliposomes. a** Time-dependent uptake of $^{14}$C-D-fructose by rGLUT5 reconstituted into liposomes prepared using brain-fraction-seven lipids (filled circles) and liposomes prepared using a synthetic brain-fraction-seven lipid mixture (empty circles). Non-specific uptake was recorded using protein-free liposomes of brain-fraction-seven (filled squares) and the synthetic-brain-seven lipid mixture (empty squares), respectively. Error bars indicate mean ± s.e.m. of $n = 3$ independent experiments. **b** Relative rGLUT5 [$^{14}$C]-D-fructose uptake after 2 min using proteoliposomes prepared from synthetic-brain-seven lipids. Individual lipids were omitted as follows: POPS, POPC, SM, POPI, Cerebroside, and POPE. Black bars represent specific rGLUT5 transport after subtraction of unspecific transport and normalized to the intensity of synthetic-brain-seven lipid mixture (synthetic BF7) where no lipids were removed. Error bars indicate mean ± s.e.m. of $n = 9$ independent experiments for synthetic BF7, -POPS, -POPC, -POPI, -Cerebroside and mean ± s.e.m. of $n = 9$ independent experiments for -SM. **c** Effect on rGLUT5 transport by titration of POPC/POPS in synthetic-brain-

seven lipid mixture proteoliposomes. $^{14}$C-D Fructose uptake was recorded after 2 min. Pie charts on top of each black bar represent the modified lipid concentrations; in black POPC, in blue POPS, and the remaining composition in clear gray. Data was represented as in (**b**), and Error bars indicate mean ± s.e.m. of n = 6 independent experiments. **d** Effect on rGLUT5 transport by titration of POC/POPE in synthetic-brain-seven proteoliposomes. As in **c**, POPE is represented in cyan. Data were represented as in (**b**); error bars indicate mean ± s.e.m. of $n = 6$ independent experiments. **e** Relative rGLUT5 $^{14}$C-D-fructose transport after 2 min in synthetic brain-fraction-seven proteoliposomes. POPC (18:1) was substituted for the greater unsaturated lipid PC (18:6) and DOPC (18:2). Error bars indicate mean ± s.e.m. of $n = 9$ independent experiments for PC (18:1) and PC (18:6) and mean ± s.e.m. of $n = 7$ independent experiments for PC (18:2). **f** Effect on rGLUT5 transport by titration of POPC/DOPC in synthetic brain-fraction-seven proteoliposomes. As in c and d, but DOPC is represented in dark orange; error bars indicate mean ± s.e.m. of $n = 6$ independent experiments.

having established the importance of membrane fluidity for GLUT transport, we can start to re-evaluate some physiological processes. In particular, rats deficient in unsaturated fatty acids have been shown to have a 30% decrease in glucose uptake into the brain, although GLUT1 and GLUT3 surface protein levels were unchanged[77]. Although free fatty acids (FFA) are naturally present in low amounts of cell membranes (0.3 to 10% total lipids), they rise to very high levels in obese patients and affect membrane bilayer fluidity[78,79]. It is well-known that increased FFA is associated with both an increase in insulin resistance and the development of type 2 diabetes[78,80]. It has been demonstrated that the rate-controlling step for FFA-induced insulin resistance in humans is the impairment of glucose transport by GLUT4[81]. In addition to external metabolic reprogramming leading to the development of type 2 diabetes in obese patients and improper GLUT4 trafficking[82], our analysis reveals that GLUT4 D-glucose uptake itself is likely to be directly impaired by FFA levels altering membrane fluidity.

In summary, by establishing mammalian GLUT kinetics and lipid composition influences in a reconstituted-liposome system, we have been able to gain an understanding of how GLUT transporters function and respond to lipids, with important physiological ramifications.

# Methods
## Materials
Lipids used were obtained from different sources for crude extracts: *E. coli* total extract (100500 P), *Bovine* total liver extract (181104 P), Soya extract (341602 G), and 1,2-dioleoyl-sn-glycero-3-phosphocholine, 18:1 (Δ9-Cis) PC (DOPC; 850375), 1,2-dilinolenoyl-sn-glycero-3-phosphocholine, 18:3 (Cis) PC (850395) and 1-palmitoyl-2-oleoyl-sn-glycero-3-phospho-L-serine, 16:0-18:1 PS (POPS; 840034) were purchased from Avanti; *Bovine* brain extract type I (B1502), *Bovine* brain extract type VII (B3635), Cholesterylhemisuccinate (C6013) and Phosphatidylcholine (PC) from soya bean (P5638) were purchased from Sigma. Ceramide galactoside (56-1051-8), Sphingomyelin (bovine brain:56-1080-11), 1,2-Dioleoyl-sn-Glycero-3-Phosphatidylethanolamine (POPE; 37-1828-9), Phosphatidylinositol (POPI; yeast, 37-0132-7) and 1-Palmitoyl-2-Oleoyl-sn-Glycero-3-Phosphatidylcholine (POPC; 37-1618-12) were purchased from Larodan.

## Construct design and cloning
*Rattus norvegicus* GLUT5 (UniProt: P43427) and *Plasmodium falciparum* Hexose Transporter 1 (UniProt: O97467) encoding genes were cloned as described previously in[33,34]. hGLUT1 (Uniprot: P11166), hGLUT3 (Uniprot: P11169), and hGLUT4 (Uniprot: P14672) encoding genes were synthesized and cloned into the mammalian expression pCDNA3.1 (+) by GeneArt. A TEV cleavage site (ENLYFQ) and eGFP (Uniprot: C5MKY7) sequence containing a twin strep tag were introduced at the C-terminal in that order.

## Large-scale production and purification
For rGLUT5 and *Pf*HT1, the protocol was followed as described previously in[33,34], respectively. For large expression of hGLUT1, 1.4 L Human embryonic kidney (HEK293F) cells (Gibco) was diluted to $0.5 \times 10^6$ cells mL$^{-1}$ with FreeStyle 293 expression medium 24 h before transfection. HEK293F cells were grown at 37 °C, 125 rpm, 8% CO$_2$, and 8% humidity. Plasmid purification was performed using PureLink Hipure Expi plasmid giga prep kit (Invitrogen). Transfection was done at a final cell density of $1 \times 10^6$ cells mL$^{-1}$. For transfection, pCDNA3-hGLUT1 plasmid was added to 140 mL of FreeStyle 293 medium to ensure later a final concentration of (0.5 µg) DNA per $1 \times 10^{-6}$ cells. PEI max (Polysciences) (Fisher Scientific) was added to a final concentration of 0.002 mg/mL and vortexed. The DNA-PEI max mix was incubated for 20 min at room temperature. The mixture was subsequently added to the culture (1400 mL), and sodium butyrate (Sigma-Aldrich) was added to a final concentration of 5 mM to enhance protein expression. The culture was incubated in a Minitron shaker (INFORS HT) at 125 rpm at 37 °C, 8% CO$_2$, and 8% humidity for 72 h. Cells were harvested by centrifugation at 3000$g$ for 5 min at room temperature. Cell pellets were resuspended in hypotonic cell resuspension buffer containing 50 mM Tris HCl pH 8.0, 80 mM Sorbitol, 50 mM NaCl, 5 mM MgCl$_2$, and 1 mM EDTA with a small amount of deoxyribonuclease I from bovine pancreas (Sigma-Aldrich) and incubated at 4 °C for 10 min with gentle agitation. Later, cells were sonicated for 8 s for 5 cycles with 30 s pause between cycles on ice. After sonication, additional sorbitol and NaCl were added to final concentrations of 520 mM and 250 mM, respectively, and the resulting solution was left at 4 °C for 10 min with mild agitation. Afterward, cell debris was removed by spinning at 3000$g$ for 10 min at 4 °C and the supernatant was ultracentrifuged for 1 h at 190,000$g$ at 4 °C. Membranes were collected, homogenized in 1× PBS pH 8.0, flash-frozen and stored at −20 °C. Membranes of hGLUT1 were diluted to 8 mg mL$^{-1}$ calculated by BCA assay and solubilized in buffer (20 mM Tris pH: 8.0, NaCl 150 mM, DDM 2% (w/v) and CHS 0.4% (w/v)) for 1 h at 4 °C with mild agitation. Solubilized membranes were spun down at 190,000$g$ for 1 h, and the supernatant was collected. The supernatant was incubated at 4 °C with 5 mL of strep-tactin XT resin (IBA Lifesciences) pre-equilibrated in (20 mM Tris pH: 8.0, NaCl 150 mM, DDM 0.003% (w/v) and CHS 0.006% (w/v)) for 3 h with mild agitation. The slurry was then poured into a gravity flow column (Bio-Rad), and the resin was washed with 10 column volumes (CV) of washing buffer (20 mM Tris HCl pH 8.0, 150 mM NaCl, DDM 0.03% (w/v) and CHS 0.006% (w/v). Then, the protein was eluted with 2 CV of 1× BXT elution buffer (IBA Lifesciences) with DDM 0.03% (w/v), and CHS 0.006% (w/v) added. Eluted protein was concentrated using a 100 kDa MW cut-off spin concentrator (Amicon Merck-Millipore) to a final concentration of ~2 mg mL$^{-1}$, flash-frozen in liquid nitrogen, and stored at −80 °C. The same procedure was followed for hGLUT3-4, but larger culture volumes were used (4–8 L) to compensate for lower

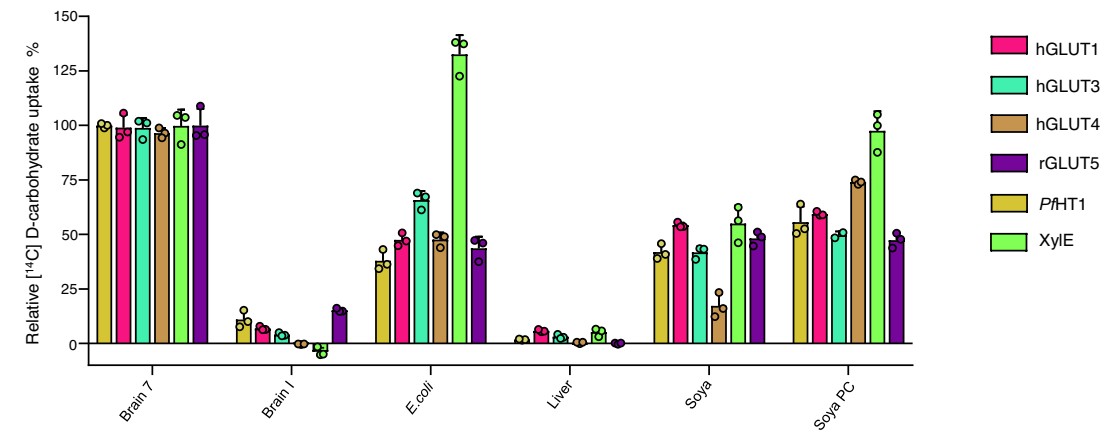

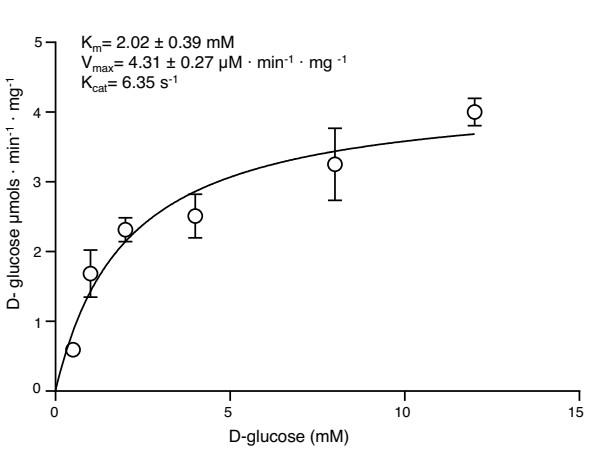

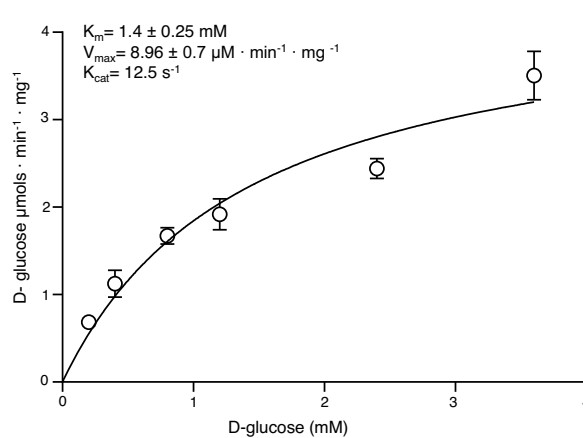

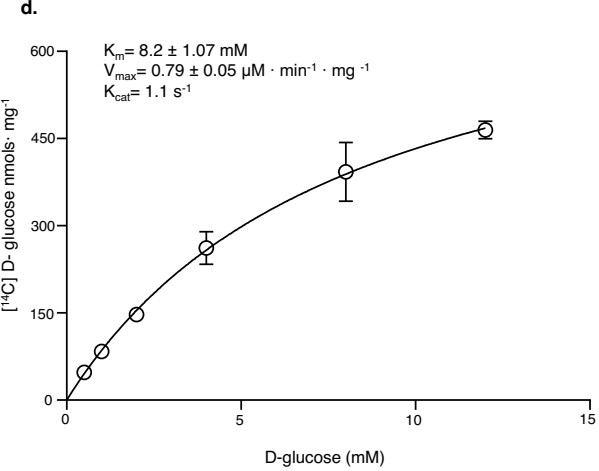

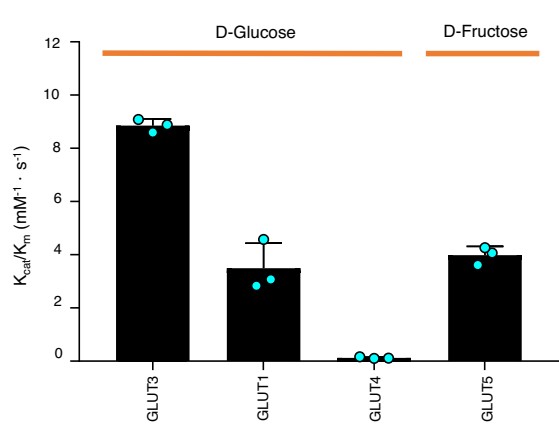

expression yields. hGLUT4 was concentrated to ~14 mg mL$^{-1}$ instead of ~2 mg mL$^{-1}$.

For XylE, large-scale expression was done using the MemStar protocol as described previously[83]. Membranes containing XylE were solubilized for 1 h at 4 °C with mild agitation in buffer containing 20 mM Tris pH 8.0, 150 mM NaCl, DDM 2.0% (w/v). The solution with solubilized membranes was centrifuged at 190,000$g$ for 1 h 4 °C. The

cleared supernatant was incubated with 20 mL of Ni$^{2+}$-nitrilotriacetate affinity resin (Ni-NTA; Qiagen) for 2 h at 4 °C with agitation after the addition of imidazole to a final concentration of 20 mM. Then slurry was poured into a gravity flow column (Bio-Rad), and the resin was washed with 20 CV of wash buffer (20 mM Tris pH 8.0, 150 mM NaCl, DDM 0.1% (w/v), 20 mM imidazole), followed by an additional wash of 20 CV with wash buffer but with a concentration of 30 mM imidazole.

**Fig. 5 | Lipid screening for human GLUT1, GLUT3, and GLUT4 mediated uptake and their corresponding *zero trans* influx kinetics. a** Relative $^{14}$C-D-glucose 30 s uptake for hGLUT1 (red bars), hGLUT3 (light-green bars), hGLUT4 (brown bars), and *Pf*HT1 (mustard bars) and $^{14}$C-D-fructose 2 min uptake for rGLUT5 (purple bars) and $^{3}$H-D-xylose 30 s uptake for XylE (lime bars) in proteoliposomes prepared using the different lipid extracts, as in Fig. 1a–d. Non-specific sugar uptake was recorded using protein-free liposomes and subtracted from the recorded specific uptake signal. The uptake was normalized to the brain-fraction-seven lipid extract. Error bars indicate mean ± s.e.m. of *n* = 3 independent experiments. **b** *Zero trans* kinetics

of D-glucose uptake for hGLUT1. All data points at their respective D-glucose concentration were measured at 50 s and fitted to a Michaelis–Menten plot with $K_{M}$, $V_{max}$, and $k_{cat}$ values subsequently calculated. Error bars indicate mean ± s.e.m. of *n* = 3 independent experiments. **c** As in Fig. 5b for hGLUT3. **d** As in Fig. 5b for hGLUT4. **e** Comparison of GLUT-specific activity ($k_{cat}/K_{M}$). Black bars represent the calculated specific activity for each GLUT; D-fructose for rGLUT5 and D-glucose for hGLUT1, hGLUT3, and hGLUT4. Data were obtained from previously calculated $K_{m}$, and $k_{cat}$ and error bar represent s.e.m of *n* = 3 independent experiments.

Subsequently, the protein was eluted with 2 CV of elution buffer (20 mM Tris pH 8.0, 150 mM NaCl, DDM 0.1% (w/v), 250 mM imidazole). The eluate was dialyzed against 3 L of a buffer consisting of 20 mM Tris HCl pH 8.0, 150 mM NaCl, DDM 0.03% (w/v) overnight at 4 °C. The dialyzed protein solution was concentrated using a 100 kDa MW cut-off spin concentrator (Amicon Merck-Millipore) to a final concentration between 2 and 5 mg mL$^{-1}$.

**Proteoliposome preparation: crude extracts**
Lipid extract and CHS in the powder were mixed in a buffer containing 10 mM Tris-HCl pH 7.5 and 2 mM MgSO$_4$ to a final concentration of 30 and 6 mg mL$^{-1}$, respectively. The resulting lipid mixture was subjected to multiple rounds of freeze–thaw cycles by flash-freezing in liquid nitrogen and thawing at room temperature interspersed with sonication. The lipid mixture was centrifuged at 16,000*g* for 15 min, and the supernatant containing small unilamellar vesicles was collected. To make proteoliposomes, (10–30 µg) of purified protein was added, with the exception of hGLUT4, where (140 µg) of purified protein was added instead to 500 µL of unilamellar vesicles, flash-frozen using liquid nitrogen and thawed at room temperature. Liposomes were stored at −80 °C after freezing and thawed when needed. After thawing them, large unilamellar proteoliposomes were prepared by extrusion (LiposoFast, Avestin; membrane pore size, 400 nm) 11 times.

**Proteoliposomes preparation: synthetic liposomes**
Individual lipids were dissolved in chloroform: methanol (2:1) at a final concentration of 50 mg mL$^{-1}$ except CHS, which was dissolved at 20 mg mL$^{-1}$ and stored at −20 °C. The individual lipids were mixed in a 25/50-mL glass round-bottom flask by stirring until the mixture was completely transparent. The solvent was removed using a rotatory evaporator (Hei-Vap Core; Heidolph) under reduced pressure of −1 atmosphere at 35 °C and 250 rpm for 30 min. The resulting lipid film was resuspended in a buffer containing 10 mM Tris-HCl pH 7.5 and 2 mM MgSO$_4$ to the same concentration as in crude extracts. In order to make sure the suspension was homogenous, stirring at 250 rpm at 35 °C for 30 min was done. An additional vortex of 5 min was done to make sure the lipid film was completely in suspension. The next following steps were the same as mentioned earlier for the crude extracts. The synthetic brain-seven mixture was prepared to match the lipid composition determined using lipidomics and was; cerebroside (30%) 9 mg mL$^{-1}$, sphingomyelin (15%) 4.5 mg mL$^{-1}$, POPC (35%) 10.5 mg mL$^{-1}$, POPE (15%) 4.5 mg mL$^{-1}$, POPI (0.5%) 0.15 mg mL$^{-1}$ and POPS (4.5%) 1.5 mg mL$^{-1}$, and in addition added CHS to a final concentration of 6 mg mL$^{-1}$. In case an individual lipid was omitted (Fig. 4c), the concentrations of the remaining lipids were increased proportionally to reach a final lipid concentration of 30 mg mL$^{-1}$.

**Protein reconstitution and orientation calculation**
To calculate protein reconstitution efficiency, freshly prepared proteliposomes (300 µL) were spun down at 250,000*g* for 45 min, and 200 µL of the resulting supernatant was carefully discarded. To the remaining liposome solution buffer (10 mM Tris-HCl pH 7.5 and 2 mM MgSO$_4$), DDM 2% (w/v) was added to a final volume of 300 µL. Proteoliposomes were solubilized at 4 °C for 1 h with mild rotation and then spun down at 250,000*g* for 45 min. Again, the supernatant was

taken out carefully, and 30 µL was run in an SDS-PAGE Tris-Glycine gel (Novex WedgeWell 4–12%; Invitrogen) at not more than 100 mV. The gel was stained with Coomassie blue (Quick Coomassie Stain; Neo-Biotech), and an image was taken using (Gel Doc EZ Imager; Bio-Rad). Analysis of gel band intensity was performed using ImageJ, and band densitometry was performed on the rGLUT5 band (Supplementary Fig. 1e). Since rGLUT5 migrates as two bands on the gel, both bands were included in the densitometry calculation. To finally calculate a relative reconstitution efficiency, the theoretical total amount of protein used for reconstitution was also analyzed in the same SDS-PAGE gel, and band densitometry was calculated as mentioned before. To estimate protein orientation in proteoliposomes, freshly prepared proteoliposomes (600 µL) were diluted into 8 mL in buffer containing 10 mM Tris-HCl pH 7.5 and 2 mM MgSO$_4$ and were spun down at 250,000*g* for 45 min. The supernatant was discarded, and the pellet was resuspended into 800 µL in 10 mM Tris-HCl pH 7.5 and 2 mM MgSO$_4$. The sample was then split into three separated conditions, 250 µL of proteoliposomes were incubated with (50 µg) of TEV and DDM 1% (w/v), 250 µL with 50 µg TEV and 250 µL without TEV and DDM. Samples were incubated overnight at 4 °C with mild agitation. Samples were later spun down at 250,000*g* for 45 min, and 100 µL of supernatant were injected into ENrich SEC 650 10 × 300 Column (Bio-Rad) pre-equilibrated with 20 mM Tris-HCl pH 7.5, 150 mM NaCl, 0.03% DDM (w/v) and run at 1 mL min$^{-1}$ using an HPLC system (Shimadzu), in the same buffer. FSEC results were analyzed using (Prism, GraphPad 7.0, and 9.5) using the area under the curve (AUC) function, and the free GFP peak was quantified. The AUC of the undigested sample was considered as background and subtracted from the digested samples with TEV; the subtraction-corrected AUC values were normalized against digested sample with TEV in the presence of DDM.

**Lipid stabilization studied by GFP-based thermal shift assay**
The thermostability of purified rGLUT5 was done as described previously[41,42]. In brief, purified GFP fusion of rGLUT5 in 20 mM Tris HCl pH 7.5, 150 mM NaCl, 0.03% DDM (w/v) purified, as described earlier, was used for thermal shift assays. rGLUT5 GFP fusion was diluted to 1 µM in a buffer containing 20 mM Tris-HCl pH 7.5, 150 mM NaCl, DDM 1% (w/v), and added the respective individual lipid at 3 mg mL$^{-1}$ followed by incubation on ice for 30 min. Afterward, β-D-Octylglucoside (Glycon) was added to a final concentration of 1% (w/v), and the samples were heated in a PCR-thermocycler (Veriti, Applied Biosystems) for 10 min at their respective temperature. Later, a 5000*g* spin for 30 min at 4 °C was done to remove aggregated protein. The supernatant was pipetted out carefully and transferred to a 96-well plate (Greiner), and fluorescence was measured excitation at 485 nm and detecting emittance at 538 nm using a FluoroSkan plate reader (Thermo Fisher Sci., SkanIt software 6.0.2). The apparent melting temperature (Tm) was calculated by plotting the mean GFP fluorescence intensity from three independent biological repeats per temperature and fitted to a sigmoidal plot using GraphPad prism.

**GLUT proteoliposome in vitro transport assays: time course and kinetics**
For single time point and time course experiments, 15 µL of proteoliposomes prepared as mentioned before were added into 45 µL of

external assay buffer (A.B) containing 10 mM Tris-HCl pH 7.5, 2 mM MgSO$_4$ with either $^{14}$C-D-glucose (30 μM), $^{14}$C -D-fructose (6.0 μM), $^{3}$H-D-xylose (0.3 μM) (American Radiolabelled Chemicals). Uptake was stopped through the removal of ligands by the addition of 1 mL of A, B, and rapid filtering through a 0.22-μm mixed cellulose hydrophilic filter (Millipore). Filters with proteoliposomes were washed individually using 6 mL of assay buffer, transferred to scintillation vials, and added 5 mL of Ultima Gold scintillation liquid (Perkin Elmer). 5 mL of Ultima Gold scintillation liquid (Perkin Elmer) was added. Radioactive decay of isotope-labeled ligands was recorded using a scintillation counter (TRI-CARB 4810TR 110 V; Perkin Elmer). For competitive-uptake assays, $^{14}$C-D-fructose uptake was measured from a sample collected 60 s after the addition of ligand, using unlabeled sugars at 50 mM in A.B.

For the estimated GLUT kinetics, the initial velocities were calculated based on the time course measurements using; for rGLUT5 (Fig. 1e) 50 s and 30 s for hGLUT1,3,4 respectively (Supplementary Fig. 4f–h). The ratio of sugars (D-fructose or D-glucose) was a mix of labeled and unlabeled sugars (~1:200), and this ratio was kept constant for the whole experiment while increasing the total sugar concentration. For each concentration of sugars tested, recorded decay values from protein-free liposomes were subtracted from the respective proteoliposomes values. Subtracted values were fitted to Michaelis–Menten kinetics, and $K_M$ and $V_{max}$ was calculated using GraphPad Prism 7.0. For counterflow XylE experiments, proteoliposomes were prepared as mentioned before but preloaded with 10 mM D-xylose in A, B. Preloaded proteoliposomes of XylE were spun down for 45 min at 250,000$g$, and the resulting supernatant carefully discarded and the collected proteoliposomes were resuspended gently to ~70 mg mL$^{-1}$ in A, B (concentrated 2.3 times). Totally, 5 μL of concentrated proteoliposomes were mixed with 45 μL of A, B without D-xylose for 2 min. The reaction was started, stopped, and the filter washed, as mentioned earlier. For the inhibition (IC$_{50}$) assays, 20 μL of proteoliposomes were mixed with 25 μL of A, B containing DMSO 5% (w/v) and 4-methylphenyl-N-(4-methylphenyl) carbamate (H4) at desired concentration for 2 min, the reaction was stopped, and decay counts from $^{14}$C-D-fructose recorded as stated before. Decay counts from empty liposomes were subtracted from the respective proteoliposome signals and normalized to the lower inhibition concentration (0.01 μM). Data were plotted using the nonlinear function one phase decay from GraphPad Prism 7.0.

## Virtual screening for rGLUT5 inhibitors

Molecular docking was carried out against the structure of rat GLUT5 (PDB ID: 4YBQ) using DOCK3.6 (https://dock.compbio.ucsf.edu/DOCK3.6/)[84,85]. Crystallographic waters and other solvent molecules were removed from the structure. Protons were added by REDUCE[86] and optimized based on visual inspection of hydrogen bonding networks. Scoring grids were based on parameters from the AMBER force field[87], and the D-fructose binding site of GLUT5 was defined based on D- glucose binding observed in a GLUT3 crystal structure (PDB ID: 4ZW9). The side chain dipole moments of key residues (166, 287, 288, 324, 418) were increased to favor protein-ligand hydrogen bonds[88]. Docking of D-fructose to GLUT5 was performed and resulted in a binding mode similar to D-glucose in GLUT3. A library with commercially available compounds (1.6 million fragment-like molecules) from the ZINC15 database (http://zinc15.docking.org) was then docked to the binding site. Based on the docking scores, 500 top-ranked molecules with a maximum of three rotatable bonds were inspected visually, and 30 of these were selected for experimental evaluation in the competitive-uptake assays (Supplementary Fig. 3a, b). Analogs of the best compound (C2) were identified based on similarity searches in commercial chemical libraries combined with molecular docking calculations.

## Mass spectrometry

For native MS analysis, 30 μM of rGLUT5 stock solution was exchanged into 100 mM ammonium acetate, pH 6.9, and 2× CMC DDM using Biospin 6 columns (Bio-Rad). Mass spectra were acquired on a Micromass LCT ToF modified for analysis of intact protein complexes (MS Vision, The Netherlands) equipped with an offline nanospray source. The capillary voltage was 1.5 kV, and the RF lens was 1.5 kV. The cone voltage was 300 V, and the pressure in the ion source was maintained at 3.0 mbar to facilitate detergent removal through collisional activation[89]. Data were analyzed using the MassLynx 4.2.

## Reporting summary

Further information on research design is available in the Nature Portfolio Reporting Summary linked to this article.

## Data availability

Data supporting the findings of this paper are available from the corresponding author upon request. The source data underlying all figures are available as a Source Data file provided with this paper. Source Data for Figs. 1, 2, 3, 5 and Supplementary Figs. 1, 3, 5, 6 are provided with this paper in the Source Data file. Compounds from the ZINC15 database can be found on [http://zinc15.docking.org]. Source data are provided in this paper.

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

## Acknowledgements
We thank the Swedish metabolic center for lipidomic analysis and Magnus Claesson for the critical reading of the paper. This work was funded by the Knut and Alice Wallenberg Foundation (D.D.), the Novo Nordisk Foundation (D.D.), the Swedish Research Council (J.C.), and the Swedish strategic research program eSSENCE (J.C.). The computations were enabled by resources provided by the Swedish National Infrastructure for Computing (SNIC) at NSC, partially funded by the Swedish Research Council through grant agreement No. 2018-05973.

## Author contributions
D.D., A.S., and A.Q. designed the project. Expression and purification of GLUT proteins were carried out by A.S., A.Q., and Y.C. Transport assays were carried out by A.S. and A.Q. Lipid analysis was carried out by A.S. and S.M. Native MS was performed by S.E.M and M.L. In silico GLUT inhibitor screening was performed by M.C., A.R., and J.C. The paper was written by D.D., A.Q., and A.S., with contributions from all authors.

## Funding

## Competing interests
The authors declare no competing interests.
