## [Peer Review File · Nature Communications]

Establishing mammalian GLUT kinetics and lipid composition influences in a reconstituted-liposome systemReviewers' Comments:

Reviewer #1:

Remarks to the Author:

This paper provides excellent characterization of a system for reconstituting functionally active GLUT5 for quantitative kinetic analyses. Using this assay system, the authors make an important contribution to understanding of the role of membrane lipids in GLUT function. They also test the ability of this system to facilitate the identification of novel small molecule GLUT inhibitors. The methodology employed is generally sound and the analysis is sufficiently rigorous to support conclusions made. While this study provides significant advancement of this field of research, there are some important considerations regarding the framing of the significance of these findings. Several minor revisions are also needed.

1. Claims that the proteoliposome system recapitulates *in vivo* function (Page 5, lines 32-34) are overstated. Several observations in the paper lead to questions whether specific lipid composition and/or purified transporter structure in this system is fully reflective of native activity. This does not diminish the value of the authors' contribution to assay optimization. The K_m is indeed similar but the turnover number is not necessarily the same (as the authors acknowledge, this is difficult to prove *in vivo*). There are also differences in GLUT5 inhibitor binding.
2. Since GLUT5 is normally expressed in intestinal epithelial cells, it is not clear why the authors did not directly test the activity of this transporter under conditions that approximate enterocyte lipid composition.
3. The title speaks of lipid "preferences". It is not clear what is meant by use of this term. It would be more appropriate to use "influences" or similar word to convey that lipid composition affects activity.
4. The final sentences of the abstract contain considerable speculation. While there are indeed striking implications of the potential relevance of these data in relation to human disease, the paper does not directly address this connection. It would be more appropriate to simply state this potential in the abstract and reserve the more speculative elements to the discussion.
5. Oligomerization of some of the GLUT isoforms is known to affect transport kinetics, but this is not considered or discussed in the analysis of this reconstituted protein assay system. Some mention of this in the discussion would be appropriate.
6. It is noted that in the sugar competition assays shown in Figs 2a and 2b, the reported relative GLUT5 activity for fructose transport with the sugars is of the same magnitude (60-80%) as some of the small molecule inhibitors reported in Figs 2d and 2e. In the former, it is concluded that these sugars do not compete for uptake, but in the later they are concluded to serve as relatively weak inhibitors. While it is recognized that the concentrations of sugars and putative inhibitors are orders of magnitude different, the basis for the different conclusions should be discussed. Is this due to a systematic dilution effect in the way the assays were conducted?
7. Figure 5d appears to have mislabeled axes. As shown, it is a time-course for GLUT4-mediated transport in reconstituted liposomes rather than a plot of glucose uptake at 50 sec vs substrate concentration as stated in the figure legend.

Minor:

1. In the introduction on page 3, line 12, GLUT1 deficiency syndrome is also known as De Vivo syndrome not De vivi
2. On page 8, line 26, reference to the lipidomic analysis should be Fig 3d not 2d.

Reviewer #2:

Remarks to the Author:

Suades et al. present a set of biochemical reconstitution experiments of the GLUT transporters to examine in details the affinities and kinetics of sugar transport. After optimizing their proteoliposome assay, they vary in a systematic manner membrane composition, using crude or synthetic lipids, to better assess the enzymology of this important class of membrane transporters. As pointed by the authors, there is fairly good structural understanding of the GLUT transport cycle, but in vitro transport assays have been unable to replicate the kinetic estimates measured from in vivo, whereas in vivo assays suffer from many, and often inaccurate approximations.

Overall, the kinetic analysis presented here attest of the robustness of their reconstitution procedure, with a sensitivity sufficiently high to determine lipid preference, mutation and isoform effects, and also inhibition by not so potent small molecules, etc. Together, these results convincingly demonstrate that GLUT proteins are very sensitive to the membrane bilayer composition and fluidity; a primordial effect is that is often overlooked when studying these transporters.

The experiments are exhaustive and carefully executed. They help to define how the GLUT transporter functions and responds to lipids, with in-turn can help understand the role of the isoforms in the physiological context. They also demonstrate the importance of carefully optimizing proteoliposome reconstitution to obtain reliable kinetics. This work will appear as a reference study in the field.

The main downside is the novelty. There are considerable earlier studies with other transporters (e.g. LacY permease, Sec translocase, ABC transporters, etc..) that have reported lipid specificity or preferentiality to gain optimal activity; as such the concept included in the manuscript is not novel. No doubt that establishing a clean GLUT reconstituted-liposome system for kinetics and lipid preferences studies is important to the field, but this makes the study rather technical.

Minor comments:

It is not clear how the authors assess their reconstitution efficiency. Section Material and Method provides "protein reconstitution calculations" but the experimental work is not well described in the section result. The Supplementary Fig.1(e) reports experimental data, but this figure should be placed in the main text and further commented. For example, rGLUT5 seems to migrate as two distinct protein species around 40 kDa.

Similarly, authors should comment on how they consider the possible reversed orientation of GLUT in the proteoliposome and how this can impact transport measurement.

Given the number of lipid species employed, I feel it would be important to systematically control the approximate structure (e.g negative stain EM) the supposed large unilamellar proteoliposomes, as well as the degree of curvature, which can potentially impact the activity of GLUT more than fluidity itself. Figure 3.d. could benefit from an embedded colour legend.

Sup Figure 1 (c). The legend is there, but the structures of potential rGLUT5 inhibitors is missing.

Establishing mammalian GLUT kinetics and lipid composition influences in a reconstituted-liposome system

We appreciate the positive response concerning our manuscript. We have carefully examined each remark and responded to all points below.

REVIEWER COMMENTS

Reviewer #1 (Remarks to the Author):

This paper provides excellent characterization of a system for reconstituting functionally active GLUT5 for quantitative kinetic analyses. Using this assay system, the authors make an important contribution to understanding of the role of membrane lipids in GLUT function. They also test the ability of this system to facilitate the identification of novel small molecule GLUT inhibitors. The methodology employed is generally sound and the analysis is sufficiently rigorous to support conclusions made. While this study provides significant advancement of this field of research, there are some important considerations regarding the framing of the significance of these findings. Several minor revisions are also needed.

Thank you for your positive feedback.

1. Claims that the proteoliposome system recapitulates in vivo function (Page 5, lines 32-34) are overstated. Several observations in the paper lead to questions whether specific lipid composition and/or purified transporter structure in this system is fully reflective of native activity. This does not diminish the value of the authors' contribution to assay optimization. The K_m is indeed similar but the turnover number is not necessarily the same (as the authors acknowledge, this is difficult to prove in vivo).

We see your point. We have clarified that proteoliposome assay is able to measure robust GLUT5 kinetics as compared to the in vivo situation. The problem is that most in vivo estimates only report a V_{max} and the turnover numbers, when reported, vary tremendously. Which numbers are the correct ones? Or, are they all compromised as there is too much background from kinase activity or suffer from difficulties in quantification of the exact GLUT transporter expressed on the cell surface? In standard biochemistry textbooks (e.g., Lehninger Principles of Biochemistry (8th edition)), only the K_M differences between GLUT transporters are discussed yet, like

any enzyme, the turnover number is just as important parameter as the K_M . This is the first time GLUT kinetics (both K_M and k_{cat}) for a number of different GLUTs have been measured with the same setup in a controlled environment absent of competition by *in vivo* metabolism and phosphorylation. While the absolute turnover in a specific cell might be different, we are very confident about the relative performance (specificity constant K_M/k_{cat}) of the different GLUT transporters reported herein. Human have 14 different GLUT transporters and even yeast have 20 different hexose transporters, why? In fact, the sugar porter family is the largest and most wide-spread MFS transporter across all kingdoms¹. The reason we believe is that D-glucose is the most important energy source and its distribution into tissues is controlled by these GLUT transporters. As such, the GLUTs have highly evolved kinetics to individually perform the D-Glucose uptake adjusted to their physiological role. This involves more than just their respective K_M , hence knowing their relative performance is very important. We have been in contact to groups modelling cancer who are excited to have better estimates for GLUT performance.

There are also differences in GLUT5 inhibitor binding.

Yes, some of the reported GLUT5 inhibitors based on *in vivo* data are indeed not inhibitors. The reported MSNBA inhibitor² does not result in any detectable inhibition in our assay. Presented below is a full IC_{50} for MSNBA addition to GLUT5 proteoliposomes, and we observe no inhibition, not even at higher concentration of 100 μ M. It is possible those inhibitor differences are a consequence of different species of GLUT5 (rat vs human), but, if we take into account that the binding site between rat and human GLUT5 is identical (and they have 81 % sequence identity overall), the fact that we do not observe even weak inhibition is surprising. Its more likely that MSNBA has an indirect, off target effect, on GLUT5, as reported from transport studies *in vivo*.

In other cases, we have been able to replicate inhibition of the reported *in vivo* inhibitors, such as those targeting PfHT1 (*Plasmodium falciparum* hexose transporter 1) e.g. MMV, 3361, and TOSH (WU-1) in proteoliposomes, and our calculated IC_{50} match *in vivo* affinities³⁻⁵. We believe *in vitro* validation should be a routine procedure for inhibitors and validation of probes and, therefore, the need for more robust *in vitro* assays.

2. Since GLUT5 is normally expressed in intestinal epithelial cells, it is not clear why the authors did not directly test the activity of this transporter under conditions that approximate enterocyte lipid composition.

We would have liked too, but a general extraction of lipids from intestinal epithelial cells are not commercially available. Moreover, an enterocyte membranes composition varies from the brush border membrane to basolateral membrane and therefore a general extraction of lipids would not be optimal to evaluate GLUT5 either. Nonetheless, from some older literature we find that the lipid composition from enterocytes^{6,7} is actually quite similar to the lipid composition of the used brain fraction 7. Interestingly, the brush border membrane has a unique fluidity level compared to basolateral membranes that was consistent to our current findings⁸.

3. The title speaks of lipid “preferences”. It is not clear what is meant by use of this term. It would be more appropriate to use “influences” or similar word to convey that lipid composition affects activity.

Thank you for the suggestion. We have used your suggestion and replaced “preferences” with “influences”.

4. The final sentences of the abstract contain considerable speculation. While there are indeed striking implications of the potential relevance of these data in relation to human disease, the paper does not directly address this connection. It would be more appropriate to simply state this potential in the abstract and reserve the more speculative elements to the discussion.

We have re-written the abstract and now the reference to metabolic disorders based on high levels of free fatty acid are more speculative.

5. Oligomerization of some of the GLUT isoforms is known to affect transport kinetics, but this is not considered or discussed in the analysis of this reconstituted protein assay system. Some mention of this in the discussion would be appropriate.

That's an interesting question. We don't see any differences between the purified oligomeric state between the various GLUT proteins and all cryo EM and crystal structures have been monomers, apart from the dimeric PfHT1 structure. We have now added a caveat in the discussion that PfHT1 kinetic estimates might be biased due to oligomerization and orientation-preferences (as later shown).

6. It is noted that in the sugar competition assays shown in Figs 2a and 2b, the reported relative GLUT5 activity for fructose transport with the sugars is of the same magnitude (60-80%) as some of the small molecule inhibitors reported in Figs 2d and 2e. In the former, it is concluded that these sugars do not compete for uptake, but in the later they are concluded to serve as relatively weak inhibitors. While it is recognized that the concentrations of sugars and putative inhibitors are orders of magnitude different, the basis for the different conclusions should be discussed. Is this due to a systematic dilution effect in the way the assays were conducted?

Thank you for this observation. In Figure 2a and 2b, radiolabeled ^{14}C -D-fructose uptake is competed for against a high concentration of different cold sugars (50 mM) since the K_M for D-fructose is 10 mM. It seems therefore reasonable to assume that the high concentration of sugars added affects sugar uptake in a non-specific manner. Consistently, in Figure 2d and e the concentration of inhibitor tested is only at 100 μM and 5% DMSO, and this low inhibitor concentration does not affect the overall signal. In order to clarify this apparent discrepancy, we included a control on figure 2d-e where glucose at a lower concentration of 100 μM plus 5% DMSO was added, as observed below, there is nearly no decrease, which we hope clarifies the differences between the two figures.

7. Figure 5d appears to have mislabeled axes. As shown, it is a time-course for GLUT4-mediated transport in reconstituted liposomes rather than a plot of glucose uptake at 50 sec vs substrate concentration as stated in the figure legend.

Thank you, this has now been corrected.

Minor:

1. In the introduction on page 3, line 12, GLUT1 deficiency syndrome is also known as De Vivo syndrome not De vivi

Thank you, this has now been corrected.

2. On page 8, line 26, reference to the lipidomic analysis should be Fig 3d not 2d.

Thank you, this has now been corrected.

Reviewer #2 (Remarks to the Author):

Suades et al. present a set of biochemical reconstitution experiments of the GLUT transporters to examine in details the affinities and kinetics of sugar transport. After optimizing their proteoliposome assay, they vary in a systematic manner membrane composition, using crude or synthetic lipids, to better assess the enzymology of this important class of membrane transporters. As pointed by the authors, there is fairly good structural understanding of the GLUT transport cycle, but in vitro transport assays have been unable to replicate the kinetic estimates measured from in vivo, whereas in vivo assays suffer from many, and often inaccurate approximations.

Overall, the kinetic analysis presented here attest of the robustness of their reconstitution procedure, with a sensitivity sufficiently high to determine lipid preference, mutation and isoform effects, and also inhibition by not so potent small molecules, etc. Together, these results convincingly demonstrate that GLUT proteins are very sensitive to the membrane bilayer composition and fluidity; a primordial effect is that is often overlooked when studying these transporters.

The experiments are exhaustive and carefully executed. They help to define how the GLUT transporter functions and responds to lipids, with in-turn can help understand the role of the isoforms in the physiological context. They also demonstrate the importance of carefully optimizing proteoliposome reconstitution to obtain reliable kinetics. This work will appear as a reference study in the field.

Thank you!

The main downside is the novelty. There are considerable earlier studies with other transporters (e.g. LacY permease, Sec translocase, ABC transporters, etc..) that have reported lipid specificity or preferentiality to gain optimal activity; as such the concept

included in the manuscript is not novel. No doubt that establishing a clean GLUT reconstituted-liposome system for kinetics and lipid preferences studies is important to the field, but this makes the study rather technical.

Thank you for raising this point. In addition to the functional insight into GLUT kinetics and lipid preferences, we think a major asset of this paper is, in fact, the technical development of the proteoliposome assay itself, which can also be applied to other transporters. This is not typically an issue for bacterial transporters, but our field has struggled to get assays to work for mammalian transporters in general and it's a very big problem. Our analysis show that this is likely as a consequence of purified mammalian transporters being less stable in detergent and because they require specific lipids for activity⁹. Indeed, almost all studies of mammalian ABC transporters only measure ATP hydrolysis and not vectorial transport, due to the difficulties in establishing robust proteoliposome assays. There are a number of high-profile deorphanization studies on SLCs linked to cancer and diabetes, and yet we still lack validation of these substrate-transporter pairing from *in vitro* assays. The EU funded project RESOLUTE aimed to deorphanize all SLCs using largely *in vivo* approaches (<https://re-solute.eu/facts>) and yet it has found this to be very challenging. This is also an issue in drug development, even if high-throughput screening can be used in the initial studies, we still need validation with *in vitro* based assays, *i.e.*, due to overlapping substrate profiles and off-target effects.

Based on our previous work we know that these GLUT transporters, like many mammalian transporters, are not very stable in detergent⁹. Since we are only measuring passive sugar flux, we cannot get around a less-than-optimal-setup by applying a driving force to drive substrate accumulation. Rather, our liposomes need to be very tight, and we need enough functional protein to detect accumulation with substrates that bind with a very weak affinity, *i.e.*, only 10 mM. Given the very tough requirements for optimizing GLUT transporters to work well in proteoliposomes, we are confident that the lipid composition and assay will also work for other mammalian SLCs. Indeed, we have further studies of SLC transporters in preparation to support this.

Minor comments:

It is not clear how the authors assess their reconstitution efficiency. Section Material and Method provides "protein reconstitution calculations" but the experimental work is not well described in the section result. The Supplementary Fig.1(e) reports experimental data, but this figure should be placed in the main text and further commented. For example, rGLUT5 seems to migrate as two distinct protein species around 40 kDa.

Thank you for raising this issue. We have included some extra details in the method section to better clarify the protein reconstitution analysis and also an additional line in results section. The two distinct species for rat GLUT5 seem to be a gel artifact (that's fairly typical for some transporters) and native MS confirms one main species.

Similarly, authors should comment on how they consider the possible reversed

orientation of GLUT in the proteoliposome and how this can impact transport measurement.

That's a valid point that we took some time to better evaluate this in our setup. The typical approach is to run an SDS-gel of the proteoliposomes to estimate the reconstitution efficiency, which is what we had done. However, if you want to estimate orientation, then you need to either incubate first with an orientation-specific probe or carry out some other orientation specific assay. Since our GLUT proteins are C-terminally tagged to GFP we can estimate orientation based on TEV cleavage efficiency. Instead of re-running the samples on SDS gels, however, we thought a more accurate approach was to run FSEC traces of the proteoliposomes so that we can confirm cleavage to the folded material. In the absence of DDM, the TEV cleaves GLUT1 in proteoliposomes with 30% efficiency and with DDM present, all of the GFP is cleaved in the presence of 10-fold excess TEV. This means that for *human* GLUT1, around 80% is orientated in the physiological direction. We repeated this assay on all the GLUTs and for three independent reconstitutions. All the GLUTs show a 70 to 80% preference for the physiological orientation. Interestingly, *Pf*H1T1 shows an 80% preference towards the reverse orientation. It could be because, unlike the GLUTs, *Pf*H1T1 is a stable homodimer which crystallized dimeric¹⁰. Nonetheless, the K_M estimates of *Pf*H1T1 for D-glucose and D-fructose match the *in vivo* estimates{Qureshi, 2020, The molecular basis for sugar import in malaria parasites} and so it is unlikely the reverse orientation has much impact on these numbers. It is possible that we do underestimate the turnover of *Pf*H1T1, yet we are not confident enough to adjust our estimates based on this difference. The most important for the work presented here is that the GLUTs all have a similar biased orientation and hence will not have an impact for our conclusions on relative performance.

Given the number of lipid species employed, I feel it would be important to systematically control the approximate structure (e.g negative stain EM) the supposed large unilamellar proteoliposomes, as well as the degree of curvature, which can potentially impact the activity of GLUT more than fluidity itself.

Thank you for your point. It is possible that the integrity of the liposomes could be affected when working with a highly fluid membrane. To check that this was not the case, we used our extreme example of proteoliposomes of rGLUT5 made with heavily unsaturated POPC 3:3. As you suggested, we have checked the samples by Cryo-EM. As shown in figure below, the proteoliposomes do not seem to be broken and the integrity/curvature has not been compromised. We did not observe any broken liposomes or unusual shapes in any area when we screened different regions of the grids. In addition, we did not observe any reconstitution differences between a regular synthetic liposomes preparation with POPC 1:0 or POPC 3:3, see below. Lane 1 to 4 are different resolubilizations of proteoliposomes in different conditions, 1; Synthetic liposomes of brain 7, 2; Synthetic liposomes without PC, 3; Synthetic liposomes without POPE and 4 Synthetic liposomes with POPC 3:3 instead of regular POPC, densitometry calculation was almost the same for all cases. We think that if the integrity of the proteoliposomes would be compromised, we would have observed poorer reconstituted protein. Overall, we believe a reconstitution calculation is sufficient.

L 1 2 3 4

Figure3.d could benefit from an embedder colour legend.

This now has been added on the figures so it is easier for the reader.

Sup Figure 1 (c). The legend is there, but the structures of potential rGLUT5 inhibitors is missing.

Fair point. We initially omitted some of them since it was a bit too crowded, but we have shown the structure of all compounds.

References

- 1 Drew, D., North, R. A., Nagarathinam, K. & Tanabe, M. Structures and General Transport Mechanisms by the Major Facilitator Superfamily (MFS). *Chem Rev* (2021). <https://doi.org:10.1021/acs.chemrev.0c00983>
- 2 George Thompson, A. M. *et al.* Discovery of a specific inhibitor of human GLUT5 by virtual screening and in vitro transport evaluation. *Sci Rep* **6**, 24240 (2016). <https://doi.org:10.1038/srep24240>
- 3 Heitmeier, M. R. *et al.* Identification of druggable small molecule antagonists of the Plasmodium falciparum hexose transporter PfHT and assessment of ligand access to the glucose permeation pathway via FLAG-mediated protein engineering. *PLoS One* **14**, e0216457 (2019). <https://doi.org:10.1371/journal.pone.0216457>
- 4 Joet, T., Eckstein-Ludwig, U., Morin, C. & Krishna, S. Validation of the hexose transporter of Plasmodium falciparum as a novel drug target. *Proc Natl Acad Sci U S A* **100**, 7476-7479 (2003). <https://doi.org:10.1073/pnas.1330865100>
- 5 Kraft, T. E. *et al.* A Novel Fluorescence Resonance Energy Transfer-Based Screen in High-Throughput Format To Identify Inhibitors of Malarial and Human Glucose Transporters. *Antimicrob Agents Chemother* **60**, 7407-7414 (2016). <https://doi.org:10.1128/AAC.00218-16>
- 6 Christiansen, K. & Carlsen, J. Microvillus membrane vesicles from pig small intestine. Purity and lipid composition. *Biochim Biophys Acta* **647**, 188-195 (1981). [https://doi.org:10.1016/0005-2736\(81\)90245-5](https://doi.org:10.1016/0005-2736(81)90245-5)
- 7 Carlsen, J., Christiansen, K. & Bro, B. Purification of microvillus membrane vesicles from pig small intestine by immunoabsorbent chromatography. *Biochim Biophys Acta* **689**, 12-20 (1982). [https://doi.org:10.1016/0005-2736\(82\)90183-3](https://doi.org:10.1016/0005-2736(82)90183-3)
- 8 Brasitus, T. A. & Schachter, D. Lipid composition and fluidity of rat enterocyte basolateral membranes. Regional differences. *Biochim Biophys Acta* **774**, 138-146 (1984). [https://doi.org:10.1016/0005-2736\(84\)90284-0](https://doi.org:10.1016/0005-2736(84)90284-0)
- 9 Nji, E., Chatzikiyriakidou, Y., Landreh, M. & Drew, D. An engineered thermal-shift screen reveals specific lipid preferences of eukaryotic and prokaryotic membrane proteins. *Nat Commun* **9**, 4253 (2018). <https://doi.org:10.1038/s41467-018-06702-3>
- 10 Qureshi, A. A. *et al.* The molecular basis for sugar import in malaria parasites. *Nature* **578**, 321-325 (2020). <https://doi.org:10.1038/s41586-020-1963-z>